# Neurocircuitry of Reward and Addiction: Potential Impact of Dopamine–Glutamate Co-release as Future Target in Substance Use Disorder

**DOI:** 10.3390/jcm8111887

**Published:** 2019-11-06

**Authors:** Zisis Bimpisidis, Åsa Wallén-Mackenzie

**Affiliations:** Department of Organismal Biology, Uppsala University, S-752 36 Uppsala, Sweden; zisis.bimpisidis@ebc.uu.se

**Keywords:** addiction, reward, transgenic mice, optogenetics, self-administration, cocaine, amphetamine

## Abstract

Dopamine–glutamate co-release is a unique property of midbrain neurons primarily located in the ventral tegmental area (VTA). Dopamine neurons of the VTA are important for behavioral regulation in response to rewarding substances, including natural rewards and addictive drugs. The impact of glutamate co-release on behaviors regulated by VTA dopamine neurons has been challenging to probe due to lack of selective methodology. However, several studies implementing conditional knockout and optogenetics technologies in transgenic mice have during the past decade pointed towards a role for glutamate co-release in multiple physiological and behavioral processes of importance to substance use and abuse. In this review, we discuss these studies to highlight findings that may be critical when considering mechanisms of importance for prevention and treatment of substance abuse.

## 1. Dopamine and Substance Use Disorder

Substance use disorder is a chronic, relapsing neuropsychiatric disease that occurs in a minority of recreational drug users [1]. In vulnerable individuals, the initial elevation of dopamine (DA) upon drug-consumption, which is believed to reflect the reinforcing effects of substances of abuse, will lead to characteristic behavioral patterns related to long-lasting alterations in the glutamatergic system [2,3,4,5]. Drug-seeking behavior and objectively reported craving in drug-addicts are both dependent on environmental cues associated with drug intake and can lead to relapse even after many years of abstinence [5,6].

All addictive substances influence the brain DA system which is implicated in reward processing, motivation and behavioral reinforcement [7,8]. The mesolimbic DA system, containing DA neurons located within the ventral tegmental area (VTA) and their projections to the nucleus accumbens (NAc) [9], has been shown to be particularly important for reward processing. DA release in the NAc, in particular in the medial (m) aspect of the shell (Sh) compartment, NAc mSh, has been implicated in the reinforcing properties of both natural and drug rewards [10]. DA modulates the activity of NAc neurons and their response to inputs coming from other limbic areas [1,11,12]. Abusive drugs increase DA levels preferentially in the NAc mSh as compared to the NAc core or the dorsal aspect of the striatum [1,13,14]. While these increases are related to the reinforcing properties of the drugs, the behavioral patterns observed in addicts after chronic drug use are related to more persistent neuroadaptations in the glutamatergic system. It has been shown that the psychostimulant cocaine rapidly induces synaptic plasticity in VTA DA neurons after acute, passive administration [15] and that persistent neuroadaptations can be observed in rats that have self-administered cocaine [16]. Unlike the VTA, synaptic plasticity in medium spiny neurons (MSNs) of the NAc is observed only after long-term cocaine use. Further, any changes in plasticity of the MSNs are contingent to a withdrawal period [17] and occur on MSNs expressing the DA receptor subtype 1 (D1R) [18]. Overall, synaptic changes upon drug administration occur in a time-dependent manner, so that the first changes take place in the VTA, and, after repeated administration, in the NAc, a phenomenon believed to reflect the long-lasting behavioral consequences of chronic drug intake [5]. Comprehensive review articles of putative mechanisms, particular characteristics and regional differences in the context of synaptic plasticity observed in animal models of drug addiction have been published elsewhere and further details are beyond the scope of this review.

## 2. DA-Glutamate Co-Release in the Mesolimbic System

### 2.1. DA Neurons in the VTA

Neurons that produce DA and release it into the extracellular space, either in the synapse or along the axon, are commonly referred to as dopaminergic neurons, or simply DA neurons. Molecularly, DA neurons are defined by the presence of the rate-limiting enzyme tyrosine hydroxylase (TH), which enables the production of DA, and the absence of enzymes that convert DA into noradrenalin and adrenalin [19]. Within the ventral midbrain, DA neurons are located in the VTA, substantia nigra *pars compacta* (SNc) and the retrorubral field. The VTA in turn is composed of several subnuclei, at least in rodents: the rostral linear nucleus (RLi), interfascicular nucleus (IF), parabrachial pigmented nucleus (PBP), paranigral nucleus (PN), parainterfascicular nucleus (PIF), and caudal linear nucleus of the raphe (CLi). Also, the rostral-most aspect of the area, which goes under different names, such as VTA rostral and rostro-medial VTA, is most often included when discussing the VTA [9,20,21,22,23] (Figure 1).

### 2.2. Vesicular Glutamate Transporters and the Concept of DA-Glutamate Co-Release

The midbrain DA system was described in the 1960s [25,26,27] and has been extensively studied since then. In contrast, the glutamatergic system remained more elusive for many years, to a large extent due to the lack of molecular markers that could enable their reliable identification within complex neuronal networks. Around year 2000 came the first reports of identification of the molecules that transport glutamate into presynaptic vesicles, and thereby, allow this amino acid to be used as a bona fide neurotransmitter ready to be released into the synapse upon neuronal activation [28,29,30,31,32]. These vesicular glutamate transporters, VGLUTs, which exist in three subtypes (VGLUT1, VGLUT2 and VGLUT3), have been found throughout the brain in various anatomical patterns, and their presence defines a neuron’s ability for exocytotic glutamate release upon neuronal depolarization. VGLUTs thereby serve as excellent molecular markers of glutamatergic neurotransmission. Together, VGLUT1 and VGLUT2 cover all classically described glutamatergic neurons. In addition, soon after their discovery, it was found that VGLUTs can be present within neurons assigned to another neurotransmitter phenotype than glutamatergic. For example, VGLUT3 was extensively detected in subtypes of serotonergic and cholinergic neurons, and VGLUT1 in inhibitory neurons (for review, see e.g., [33,34]). The discovery of the VGLUTs thereby opened up for a completely new view of many neuronal systems, as they could be shown to possess the molecular machinery for co-release of glutamate parallel to release of its “first” neurotransmitter, often referred to as the “primary” neurotransmitter. Glutamate co-release is a striking neuronal feature which can help explain some complex physiological features that could not be fully accounted for by the primary neurotransmitter. For example, fast excitatory neurotransmission originating from midbrain DA neurons had been recorded in striatal brain slices and mesencephalic cell cultures using electrophysiological setups, however, it was unknown how DA neurons could give rise to this type of signaling [35,36,37,38,39]. With the finding that some DA neurons contain VGLUT2, originally described in in-vitro cell-based systems [40], and subsequently confirmed in a number of histological studies as described below, a molecular basis for the electrophysiologically measurable fast excitatory post-synaptic currents could be initiated and formed.

Neurons releasing more than one neurotransmitter have been given different names to describe their complexity. Based on the capacity of certain midbrain DA neurons to co-release glutamate, a property in which glutamate is released by a neuron upon depolarization, these neurons have been referred to as co-releasing, but also “bi-lingual”, “combinatorial” and “dual-signaling” [19,22,33,34], alluding to their ability to “speak different languages”. The ability of neurons to release two or more neurotransmitters has also been referred to as “multiplexed neurochemical signaling”, or “multiplexed neurotransmission” [41]. In terms of function, the sorting of vesicular neurotransmitter transporters to subcellular domains will account for the inherent property of co-releasing two or more neurotransmitters which can occur from the same synaptic vesicle, or the same pool of synaptic vesicle within an axon, or from distinct sets of synaptic pools located in different subdomains within axons (reviewed in [33]). The kind of synaptic mechanisms that any given type of co-releasing neuron utilizes needs to be defined experimentally, which can be challenging. While “co-release” generally refers to the release of two or more neurotransmitters from any given neuron, the concept of “co-transmission”, which is often used to describe similar phenomena, has been defined more strictly: “*Co-transmission in the strictest sense implies that two neurotransmitters are released at the same time from a common pool of synaptic vesicles within one axon terminal.*” [33]. In the context of glutamate co-release from midbrain DA neurons, this type of more narrowly defined “co-transmission” from a common pool of synaptic vesicles within one axon terminal remains to be experimentally identified and defined. Instead, current data rather point towards the co-release of DA and glutamate from different microdomains in axonal terminals. With their discovery that VGLUT2 and VMAT2, the vesicular monoamine transporter, sort to distinct subpopulations of synaptic vesicles within a subset of mesoaccumbens axons in rodents, Zhang and colleagues recently concluded that “*… our results do not support the hypothesis that axon terminals from these neurons co-release dopamine and glutamate from identical axonal terminals. Rather, our findings indicate that synaptic vesicles that release dopamine or glutamate from mesoaccumbens terminals in both adult rats and adult mice are located in distinct microdomains*” [42]. Based on existing data and current terminology, we have used the term “co-release” throughout this review to broadly describe the concept of DA and glutamate release from the same midbrain neuron without specifying signaling mechanisms.

In summary, while the midbrain DA system has been the focus of attention in the field of reward and addiction for many years, the “subdiscipline” of DA–glutamate co-release is considerably younger. Consequently, the putative role of DA–glutamate co-release in neurocircuitry and behavioral regulation has remained rather unexplored. However, several studies using experimental animals point towards an impact of DA–glutamate co-release in mechanisms of relevance for addiction, suggesting that glutamate co-release is worthwhile to explore for the benefit of new prevention and/or intervention strategies for substance use disorder. Several recent reviews describe DA–glutamate co-release from different research angles, including possible packaging/release mechanisms, putative role of VGLUTs in promoting vesicular packaging of the primary neurotransmitter, post-synaptic effects of the co-released glutamate and more (see e.g., [34,43,44,45]). In this review, we will focus our attention on behavioral regulation putatively mediated by DA–glutamate co-release as discovered using rodent models. The feature of DA–glutamate co-release may contribute to dopaminergic function in reward mechanisms and may thus be of importance when considering physiological and behavioral consequences of substance use and abuse.

### 2.3. Expression Patterns of VGLUT2 in the Midbrain DA System and Validation of Glutamate Co-Release

To begin to understand functional implications of DA–glutamate co-release, it is important to know where the neurons that possess this ability are located, as only then can neurocircuitry and behavioral roles be fully delineated. The presence of VGLUT2 within midbrain areas where DA neurons reside have been described in several studies, most in which in-situ hybridization for VGLUT2 mRNA has been combined with the detection of either TH mRNA or TH protein for the visualization of DA neurons. Analyses of mouse and rat midbrains have shown that VGLUT2 mRNA-positive cells are scattered throughout the VTA and the adjacently located SNc with more frequent appearance of VGLUT2/TH co-localizing neurons medially than laterally, primarily in the medial aspect of the VTA [46,47,48]. VGLUT2-positive neurons in the ventral midbrain have also been described in primates, including humans [49]. In rodents, neurons expressing both the VGLUT2 and TH genes comprise a minority of the total number of cells in the adult VTA expressing either the VGLUT2 or the TH gene [47,50,51]. In two recent studies, we could confirm these previous analyses by showing that the highest density of VGLUT2 neurons in the VTA was found in the RLi, followed by the PBP, PN and IF. In the RLi and within a small spatially-restricted area within the PBP, barely any TH-positive neurons can be found, but here VGLUT2 is high. We have called this VGLUT2-dense area within the PBP “the subzone of the PBP” (szPBP) to distinguish it from the remaining PBP which contains VGLUT2 at lower density but TH at higher density [23,24] (Figure 1). Thus, the PBP in general has a TH^high^/VGLUT2^low^ profile, but the szPBP and RLi have the opposite profile, TH^low^/VGLUT2^high.^

VGLUT2-TH co-expressing neurons are generally sparse in adulthood, and a temporal regulation between birth and adulthood has been shown [52,53]. When addressing embryogenesis, we could readily identify VGLUT2-TH co-localization already at E12.5 [54] (Figure 2). To address the spatio-temporal profile of VGLUT2 in midbrain DA neurons in more detail, we recently performed a time-study including E14.5, newborn (postnatal day 3, P3) and adult mice (Figure 3) [24]. By co-localizing VGLUT2 mRNA with both TH and dopamine transporter (DAT) mRNAs, we found an interesting temporally dynamic pattern of expression. Almost no co-localization between VGLUT2 with either TH or DAT was detected at E14.5 (Figure 3A–C), while substantial co-localization was observed in the newborn mouse (Figure 3D–F). At P3, VGLUT2 mRNA showed prominent co-localization with both TH and DAT mRNA with higher density of VGLUT2/TH than VGLUT2/DAT double-positive neurons. This is explained by the lower expression of DAT than TH in medial aspects of the VTA, where VGLUT2 is at its highest. In addition, subareas within the VTA showed different amounts of co-localization. Primarily the PBP, but also the PN and IF showed co-localization of VGLUT2 with TH and DAT, respectively. The RLi, which shows the highest levels of VGLUT2 in the VTA, was almost devoid of co-localization of VGLUT2 with either TH or DAT due to the low abundance of these transcripts in this brain nucleus. While readily detected in the newborn mouse, the level of VGLUT2/TH and VGLUT2/DAT double-positive neurons was overall low in all VTA subareas in the adult mouse (Figure 3H,I), confirming previous studies.

To further the understanding of VGLUT2 expression in developing DA neurons, we have recently addressed VGLUT2 in DA neurons from the time-point around mid-gestation when these neurons are born and can now show that most early differentiating midbrain DA neurons express VGLUT2 at stages E10–11 (Dumas and Wallén-Mackenzie, in press, 2019). Further, we show that this early and abundant VGLUT2-expression is subsequently down-regulated as embryonal development proceeds, providing an explanation for the higher appearance of VGLUT2 in E12.5 than E14.5 described above. By including this novel piece of data into a concept of VGLUT2 expression in midbrain DA neurons, it seems that most, if not all, DA neurons initially express VGLUT2 which is subsequently downregulated during embryonal development to be upregulated in subsets of VTA neurons around birth and again down-regulated in adulthood.

In summary, VGLUT2 co-localizes to a higher extent with TH and DAT in newborn than in adult mice. With the expression of VGLUT2 in DA neurons primarily during early embryogenesis and during a subsequent phase around birth, it is interesting to speculate around putative roles of VGLUT2 in the developing DA neuron. In addition to its temporally regulated expression during the normal life span starting from embryogenesis, it has been shown that VGLUT2 expression levels can be induced in mature organisms upon stress and injury [53,55], suggesting that a glutamate co-releasing phenotype can be acquired, or at least accentuated, at different time-points during life in response to particular experiences. VGLUT2 thus shows an interesting spatio-temporal regulation pattern that may have important implications for behavioral regulation and disorders of the DA system throughout life. Below, we will discuss conditional knockout (cKO) studies in mice that support this observation.

In terms of projections of DA–glutamate co-releasing neurons, DA neurons in the medial aspect of the VTA, where VGLUT2 levels are highest, are known to project to the NAc mSh [10]. Indeed, optogenetics-driven analyses have confirmed previous observations from electrical stimulations in slice and cell culture systems and further shown that that stimulation of DA neurons in the VTA gives rise to excitatory post-synaptic currents in MSNs primarily on the ventral, rather than the dorsal, striatum [56,57]. Perhaps most attention in the DA–glutamate field has been given to MSNs, but also other striatal neurons have been shown to receive input from DA–glutamate co-releasing neurons. Lately, Chuhma and colleagues demonstrated that midbrain DA neurons induce post-synaptic glutamatergic effects in all neuronal types present in the striatum (MSNs, cholinergic interneurons (ChIs also referred as CINs) and fast-spiking interneurons (FSI)), but that the effects are different depending on the anatomical region where the neurons are located, and, specifically for the NAc mSh, most profound on ChIs [58]. While most prominently projecting to the ventral striatum, post-synaptic glutamatergic effects by DA–glutamate co-releasing VTA neurons have also been observed in these same neuronal cell types in the dorsal striatum. Interestingly, ChIs in the dorsomedial parts were shown to be affected differently than those located in the dorsolateral parts [59,60].

Having summarized current knowledge of anatomical positions of DA–glutamate co-releasing neurons in the VTA and their striatal target neurons, the next section will deal with behavioral consequences upon experimentally-achieved disruption of DA–glutamate co-release using transgenics-based approaches in mice.

### 2.4. Behavioral Consequences upon Disrupted Dopamine–Glutamate Co-Release in Transgenic Mice

The current literature contains several studies in which conditional knockout technology in mice has been used to understand if targeted disruption of VGLUT2 gene expression within DA neurons has any measurable effect on behavioral output. These studies, which are all different implementations of the Cre-LoxP system with the aim to delete glutamate co-release in DA neurons, have in common that they identify significant alterations in responsiveness to natural rewards and/or drugs of abuse (listed in Table 1). Together, these studies strongly implicate the importance of DA–glutamate co-release in mechanisms of relevance to addiction. We will discuss some of these studies in more detail as specific features might be of particular importance when considering DA–glutamate co-release in mechanisms of addiction. We will also go through putative caveats that might be important to bear in mind when interpreting these behavioral data, some of which are based on recent discoveries of so called “off-target” effects in transgenics methodology.

### 2.5. DAT-Cre-Mediated Gene Targeting of VGLUT2 in Studies Aiming to Unravel the Importance of Glutamate Co-Release in Neurocircuitry of Reward and Addiction

Parallel to findings of VGLUT2 gene expression in midbrain DA neurons, initially detected in the adult rodent VTA [46] and in DA cell cultures [40], our own histological studies identified VGLUT2 in TH-positive neurons of the ventral midbrain of the mouse not only in adulthood but also in the developing embryo already at gestational day 12.5 (E12.5) [54] (Figure 2). Such an early developmental expression suggested to us that VGLUT2 might be important for proper DA cell development, and consequently, for dopaminergic functions at adulthood. At that time, we took two Cre-LoxP approaches to disrupt VGLUT2 in midbrain DA neurons to address if DA–glutamate co-release had any influence on dopaminergic functions. In both approaches, a floxed VGLUT2 mouse line [66,67] was crossed with a Cre-driver transgenic mouse to direct the targeting event to DA neurons: (i) TH-Cre and (ii) DAT-Cre. Both these Cre-drivers are active during embryonal development, from stages around mid-gestation of the mouse embryo. We also tested the “opposite” strategy, to delete the Vesicular monoamine transporter (VMAT2) gene in cells positive for VGLUT2 using a VGLUT2-Cre transgenic mouse line to target a floxed VMAT2 allele. However, this strategy, in which we aimed to delete DA signaling rather than glutamatergic signaling from DA–glutamate co-releasing neurons, resulted in dead pups around birth. We speculated that the neonatal death was due to a breathing phenotype similar to what we observed when deleting VGLUT2 in all cells [66], but caused by monoaminergic rather than glutamatergic loss of function in the breathing circuitry. However, beyond VGLUT2 in midbrain DA neurons, VGLUT2 is expressed in additional monoaminergic neuronal populations in the medulla, including noradrenergic neurons of nuclei A1 and A2 and adrenergic neurons of C1, C2, and C3 nuclei (see e.g., [68,69]), and loss of VMAT2 in these neurons could cause the loss of a series of vital autonomic functions. Further investigations would have been required to fully address the mechanisms for neonatal lethality of targeted deletion of VMAT2 in VGLUT2-neurons, however, we did not perform such studies.

By focusing on gene-targeting of VGLUT2, using the TH-Cre approach to disrupt VGLUT2 expression, we realized early on that TH-Cre will direct targeting to VGLUT2-positive neurons beyond DA neurons, due to an early and non-monoaminergic-selective phase of TH promoter activity [70]. The *Vglut2^f/f;TH-Cre^* cKO mice showed interesting hippocampal phenotypes [70], but were not used in our laboratory for further analysis of DA–glutamate co-releasing neurons. Instead, the approach using DAT-Cre to direct the targeted deletion of VGLUT2 to DA neurons was more promising. *Vglut2^f/f;DAT-Cre^* cKO and control mice were analyzed in several behavioral, electrochemical and biochemical parameters [54]. We found that basal motor and memory functions were normal in these cKO mice, but that their risk-taking behavior was altered. We also found that in both home-cage and novel environments, *Vglut2^f/f;DAT-Cre^* cKO mice showed a greatly blunted overall response to the psychostimulant amphetamine, which exerts strong effects on DA release. Specifically interesting, in a home-cage environment in which all movements were automatically registered, cKO mice showed a strikingly different response than the control with lower total activity at several different doses analyzed (Figure 4). When analyzing horizontal and vertical locomotion parameters separately, it was apparent that the cKO and control mice had strikingly different dose-response curves in terms of behavioral output. At higher doses, locomotion and rearing decreased in the control mice as stereotypic behavior became dominating. Stereotypy was detected as bodily shakings, and in the automated recording of any movement displayed by the mice, these shakings were recorded and contributed to the parameter of total activity. In control mice, the onset of stereotypic behavior thus caused a high level of total activity despite a reduction in locomotion and rearing. In contrast to control mice, cKO mice showed less stereotypy but increased both locomotion and rearing with higher doses. Since both motor function and basal locomotor activity were normal in the cKO mice, these parameters did not contribute to the differential response observed between genotypes. The finding rather suggested that the induction of stereotypic behavior seen at the higher doses in control mice did not at all develop to the same extent in *Vglut2^f/f;DAT-Cre^* cKO mice. The absence of stereotypy likely contributed to the overall lower activity of the cKO mice compared to control mice despite the increase in locomotion and rearing displayed by cKO mice at higher amphetamine doses. In contrast to this remarkable response at higher doses, both locomotion and rearing parameters were lower in cKO than control mice at lower doses, a blunted type of locomotion which might reflect a lower release level of mesostriatal DA in the absence of VGLUT2 in midbrain DA neurons, or which might be directly caused by the reduction of VGLUT2-mediated glutamatergic neurotransmission in the circuitry. In this context, it might be important to bear in mind that VGLUT2 in DA neurons is most abundant during embryonal development (as discussed above), when the DAT-Cre-mediated targeting of VGLUT2 is initiated in the *Vglut2^f/f;DAT-Cre^* cKO mice, suggesting that behavioral manifestations displayed by the cKO mice might be due to neurocircuitry compensations. Developmental adaptations will be further discussed below. In summary, the results of this first study implementing DAT-Cre to disrupt VGLUT2 in DA neurons with the aim to probe the putative importance of DA-glutamate co-release, demonstrated a strikingly altered response curve to amphetamine in the absence of VGLUT2. This finding led us to suggest that VGLUT2 in DAT-Cre neurons is important for the behavioral response to the psychostimulant amphetamine [54]. As described below, this initial finding was substantiated with subsequent studies using additional types of rewards and behavioral paradigms.

Using mouse genetics, several different DAT-Cre transgenic lines and floxed VGLUT2 lines have been produced and combined to create various *Vglut2^f/f;DAT-Cre^* cKO lines for the study of DA–glutamate co-release. By using different versions of DAT-Cre and floxed VGLUT2 mouse lines, the different *Vglut2^f/f;DAT-Cre^* cKO lines that exist are not identical but similar, a feature which could be of importance when analyzing the results generated. Also, experimental procedures can vary between laboratories. However, while some discrepancies can been detected when comparing results from different *Vglut2^f/f;DAT-Cre^* cKO mouse lines and laboratories, overall, the findings point towards the same conclusion which implies DA–glutamate co-release in mechanisms of relevance to addiction. In one important study, Hnasko and colleagues confirmed that midbrain cell cultures from one *Vglut2^f/f;DAT-Cre^* cKO mouse line lacked VGLUT2 in synaptic boutons of DA neurons and that cKO mice had reduced excitatory post-synaptic potentials (EPSCs) on MSNs of the NAc in response to electrical stimulation of the VTA [63]. Using optogenetic stimulation of DA neurons, Stuber and colleagues subsequently demonstrated that the EPSCs on NAc MSNs were completely abolished in these *Vglut2^f/f;DAT-Cre^* cKO mice [57]. Behaviorally, *Vglut2^f/f;DAT-Cre^* cKO mice were confirmed to not have deficits in spontaneous locomotion or motor coordination [63], however, one study has reported reductions in spontaneous activity and disturbances in motor coordination in the rotarod motor test in one line of *Vglut2^f/f;DAT-Cre^* cKO mice [62]. Several studies have by now confirmed that *Vglut2^f/f;DAT-Cre^* cKO mice have blunted locomotor activity in response to acutely administered cocaine and amphetamine [54,62,63], but it has also been shown that cKO mice have normal behavioral sensitization after repeated injections of cocaine compared to their control littermates [63]. The cocaine-induced locomotor responses seem to be dissociated from the rewarding effects of cocaine since the cKO mice showed intact conditioned place preference (CPP) to the drug [63].

Following up on these initial studies in which the experimenter delivered drug injections to the rodent, we wished to advance the knowledge further by using operant self-administration methodology to come closer to the situation where the subject itself has the possibility to choose whether or not to administer a rewarding substance. Using the same *Vglut2^f/f;DAT-Cre^* cKO mouse line as in our previous experiments [54], we next analyzed the responses to both natural (sugar) and drug reward in such a self-administration paradigm to investigate how disruption of glutamate co-release from DA neurons is involved in reward processing. In the sugar self-administration test, in which nose-poking in an active nose-poke led to sucrose pellet delivery, *Vglut2^f/f;DAT-Cre^* cKO mice acquired the operant behavior similarly to control mice, but they self-administered significantly more sugar compared to controls. This was particularly evident when the task requirements were higher (FR5: 5 nose-pokes to receive a sucrose pellet) [61]. These responses were selective towards the highly palatable food and resistant to satiety; the cKO mice still consumed more calories from sucrose during the high-requirement task (FR5) compared to littermate controls [61]. When we then tested *Vglut2^f/f;DAT-Cre^* cKO mice in a cocaine self-administration paradigm, the mice displayed increased behavioral responding to receive lower doses of cocaine (Figure 5). Furthermore, during a cue-induced reinstatement phase, where nose-poking would lead only to presentation of cues associated with cocaine intake but not cocaine itself, cKO mice responded significantly more than controls just to receive the cues [61]. Taken together, these results demonstrate that mice lacking the ability to co-release glutamate from DA neurons display distinct patterns of behavior related to reward processing. While *Vglut2^f/f;DAT-Cre^* cKO mice show reduced locomotor activity in response to psychostimulants, specific aspects of the rewarding process associated to these drugs are intact. The animals form normal pavlovian associations in response to cocaine in CPP and they acquire similar rates of cocaine self-administration to control mice. In contrast, *Vglut2^f/f;DAT-Cre^* cKO mice seemed to be more sensitive to sugar and lower doses of cocaine, and strikingly, they were more susceptible to environmental stimuli associated with drug intake as shown by increased response in a cue-induced reinstatement phase during a cocaine self-administration paradigm.

To address molecular mechanisms, we next implemented a series of biochemical analyses. We found distinct molecular alterations in certain anatomical areas within the reward system of *Vglut2^f/f;DAT-Cre^* cKO mice. Interestingly, these alterations could be observed both under baseline conditions and after cocaine administration and might explain the observed behavioral patterns discussed above. *Vglut2^f/f;DAT-Cre^* cKO mice displayed elevated numbers of D1 receptors (D1R) in dorsal striatal areas and D2 receptors (D2R) in the shell region of the NAc, detected as increased binding of radio-labeled D1R and D2R ligands, respectively [61]. Furthermore, we found that cKO mice showed elevated mRNA levels of the nuclear receptor and immediate early gene Nur77 under baseline conditions in NAc core and dorsal striatal areas. The same elevation was seen in levels of c-Fos mRNA, another immediate early gene commonly used to assess neuronal activation. The elevated levels of Nur77 under baseline conditions in cKO mice were comparable to the ones observed is control mice after cocaine administration (Figure 6) [61]. Nurr77 expression is thought to be tonically inhibited by DA in the striatum and its upregulation to be related with neuroadaptations induced by chronic administration of drugs of abuse [71]. It is possible that the enhanced presence of D2R in the NAc disrupts the DA-mediated tone on Nur77 expression. Noteworthy, the detection of elevated levels of D2R might also reflect an increase in autoreceptors in presynaptic terminals related to decreased DA release. Increased levels of striatal DA receptors and immediate early genes Nur77 and c-Fos might render *Vglut2^f/f;DAT-Cre^* cKO mice more susceptible to neuroadaptations related to chronic administration of drugs of abuse, or more sensitive to their rewarding properties. This could also be the case for natural rewards. In addition, several studies have identified reduced stimulation-induced DA levels in the striatum of these mice [61,62,63]. This reduction could be related to disrupted development of the DA system [62] or due to altered intracellular dynamics and reduced synaptic packaging of DA, the “primary” neurotransmitter, due to the absence of VGLUT2 in presynaptic terminals [63,72]. In both scenarios, less DA in the synaptic cleft would result in the cessation of the inhibitory signal to Nur77 expression which in turn may lead to the long-lasting neuroadaptations and behavioral manifestations that have been demonstrated in the studies discussed above. Together, these biochemical analyses demonstrated that gene-targeting of VGLUT2 in DAT-Cre neurons leads to molecular neuroadaptations that are similar to changes observed in control animals only upon systemic administration of addictive drugs (here demonstrated with cocaine). *Vglut2^f/f;DAT-Cre^* cKO mice showed baseline levels of Nur77 and c-Fos elevated to the extent that cocaine administration did not increase them further, instead, baseline expression was already at the level observed in control mice upon drug administration. Together, the analyses of D1R and D2R availability and levels of immediate early gene expression strongly argue for a potent role of VGLUT2 in maintaining molecular homeostasis: When VGLUT2 is removed in DAT-Cre neurons from embryogenesis, there is a shift in the abundance of a range of molecular players that are important for normal functions of the DA system and this molecular shift is, in turn, likely to contribute to the behavioral manifestations observed in various reward-related experimental paradigms in which the response of the cKO mice differ significantly from that of the control mice.

Drug and natural rewards can have several sites of action and influence systems that are not directly related to DA. Experimentally, optogenetic stimulation of the DA system has been used to isolate and dissect out the role of DA neurons in reward responses [73,74,75]. Previous studies have demonstrated that optogenetic stimulation of VTA DA neurons in mice has potent reinforcing effects on behavior and can induce molecular and behavioral adaptations similar to those observed after cocaine self-administration [76]. In a recent study, *Vglut2^f/f;DAT-Cre^* cKO mice were addressed in a collaborative effort to further the understanding of DA-glutamate co-release by using optogenetic approaches to avoid non-selective effects of drug or natural rewards. In this study by Wang et al., we could confirm the reduction of glutamatergic neurotransmission in *Vglut2^f/f;DAT-Cre^* cKO mice using both patch-clamp electrophysiology in slice preparations and in-vivo amperometry in the living mouse [65] (see Viereckel et al., [77], for protocol for optogenetics-coupled glutamate electrochemistry in-vivo). By implementing an optogenetics-based intracranial self-stimulation behavioral paradigm, it was found that the acquisition of operant behavior to optogenetically self-stimulate DA neurons in the VTA remained intact in *Vglut2^f/f;DAT-Cre^* cKO mice. However, the rate of responding when parameters of stimulation changed (intensity of stimulation from 8 mW to 32 mW) was lower in *Vglut2^f/f;DAT-Cre^* cKO mice compared to littermate controls [65]. Further, cKO and control mice showed the same level of real-time place preference (RT-PP) for a compartment paired to optogenetic stimulation [65].

To further explore how gene-targeting of VGLUT2 in DAT-Cre neurons might influence dopaminergic function, we recently performed a small pilot study in which *Vglut2^f/f;DAT-Cre^* cKO and control mice were compared in an extended version of the optogenetic self-stimulation paradigm described by Wang et al. [65]. We increased the testing time compared to the previous study [65] and also added several new phases to the program in an attempt to model different aspects of food or drug self-administration: Both fixed and progressive ratios were analyzed, followed through with sessions of forced abstinence, reinstatement, extinction and cue-induced reinstatement, respectively (Figure 7A). Upon stereotaxic surgery to deliver optogenetic DNA constructs to the VTA, mice were first validated in the optogenetic RT-PP paradigm which confirmed the strong RT-PP displayed by both *Vglut2^f/f;DAT-Cre^* cKO and control mice, in accordance with the previous study [65]. Next, the mice were tested in the extended optogenetic self-stimulation program and we could again confirm that cKO and control mice showed the same acquisition rate of the self-stimulation behavior [65]. Further, our extended program found no differences between cKO and control mice upon testing in a single progressive ratio session, in a reinstatement session after a period of forced abstinence or in a 5-day extinction phase where lever-pressing no longer resulted in optogenetic stimulation or cue presentation. However, when the mice were analyzed in a cue-induced reinstatement session, where lever-pressing resulted in cue-light presentation but not laser-stimulation (Figure 7B), cKO mice responded by lever-pressing significantly more vigorously than their littermate controls (Figure 7C & D). While this extended version of the optogenetic self-stimulation program has only been addressed in a modest number of animals, the striking difference observed between cKO and control mice in their behavioral response to cue-presentation should be of interest to address further to fully validate the findings and find out any underlying mechanims.

In summary, taken together with the previous study [65], the presented results here show that the actual acquisition of operant responding in the optogenetics-based self-stimulation paradigm is not different in *Vglut2^f/f;DAT-Cre^* cKO mice compared to control mice. Beyond this observation, by covering several additional behavioral parameters, our pilot study demonstrated that targeted removal of VGLUT2 in DAT-Cre neurons is sufficient to induce profound changes in the animal’s response to cues associated with reward delivery. When considering this new result in the context of our previous data pin-pointing that the same transgenic line of *Vglut2^f/f;DAT-Cre^* cKO mice displayed increased behavioral response to cues associated with cocaine, an interesting picture appears: Mice lacking the ability for DA–glutamate co-release demonstrate distinct behavioral disturbances related to increased sensitivity to reward-associated cues. This finding strongly implies a role for DA–glutamate co-release in cue-induced reinstatement which, given its importance for relapse in human addicts, should be of particular interest to explore further in the context of interventive strategies.

### 2.6. DAT-Cre-Mediated Gene Targeting of Phosphate-Activated Glutaminase (GLS1)

As mentioned above, knocking out VGLUT2 in DA neurons during embryonal development caused reduced release of DA in striatal areas in adulthood [61,62,63]. This finding might be either the result of disturbances in the development of the DA system when VGLUT2 is no longer present in these neurons [62], due to changes in the presynaptic milieu as a result of VGLUT2 absence [63,72], or both. Mingote and colleagues [64] applied a genetic approach to reduce glutamate release from DA neurons independently of VGLUT2 with the idea to circumvent the effects on DA availability upon VGLUT2 targeting. By using a *DAT^IREScre/+^::GLS1lox/+* mouse line in which the expression of phosphate-activated glutaminase (GLS1) was disrupted on the heterozygotic level in DAT-Cre neurons, a conditional reduction of glutamate synthesis in DA neurons was achieved. Using this approach, glutamate release from DA neurons was targeted in a frequency-dependent manner; the maintenance of glutamate release was disturbed mainly during high frequencies, normally associated with reward processing. Behaviorally, mice with reduced glutamate synthesis in DA neurons exhibit attenuated amphetamine sensitization and potentiated latent inhibition [64]. Latent inhibition refers to the phenomenon during which animals display attenuated conditioning when they are repeatedly pre-exposed to the conditioned stimulus in neutral settings, a condition thought to model the inability of patients with schizophrenia to show selective attention and to ignore irrelevant stimuli [78]. Attenuated behavioral sensitization and potentiated latent inhibition were previously associated to a mouse line with reduction of glutamate synthesis throughout the brain [78], but Mingote and colleagues narrowed down their observations to prove that the effects are mediated by reduction of glutamate synthesis selectively from DAT-Cre neurons. Indeed, a mouse line with reductions of glutamate synthesis in forebrain areas did not demonstrate the same behavioral manifestations [64]. The role of DA–glutamate co-release in latent inhibition should be of interest to further research in the context of substance use/abuse.

### 2.7. Some Caveats in the Implementation of Transgenics to Address Neuronal Function

All studies discussed above implemented transgenic approaches that target glutamate co-release from early developmental stages with the common feature of DAT-Cre transgenic mouse lines to drive the recombination of floxed alleles. Since the activity of the endogenous promoters for VGLUT2 [54,70] and DAT [79] both have embryonal onset, it is possible that the observed phenotypes are the result of developmental adaptations resulting from the gene-targeting event of VGLUT2. This possibility has been discussed in the literature cited above, and indeed, it has also been experimentally shown in one *Vglut2^f/f;DAT-Cre^* mouse line that cKO mice have impaired DA neuron development; the number of midbrain DA neurons and their projections to striatal areas are significantly reduced when VGLUT2 is not present in DAT-Cre neurons [62]. Also, as discussed above, major molecular neuroadaptations occur as a consequence of VGLUT2 gene-targeting with the elevation of D1R and D2R availability and enhanced c-Fos and Nur77 expression levels at baseline conditions. Based on all of these observations, it is highly conceivable that VGLUT2 plays a developmental role in the establishment of the nervous system, a role that stretches beyond its role as pre-synaptic transporter of glutamate into synaptic vesicles. While it has been challenging to dissociate putative developmental adaptations following gene-targeting of VGLUT2 in DA neurons from effects that are solely dependent of loss of vesicular glutamate packaging, issues related to VGLUT2 were avoided using the GLS1-approach presented by Mingote and colleagues [64]. However, not only regulation of VGLUT2 but also the spatio-temporal regulation of the DAT promoter, the sequence of which is used to drive the expression of Cre recombinase in DAT-Cre mice, may also present challenges. Implementation of developmentally regulated promotors always carries the risk of neuroadaptations that might be responsible for behavioral phenotypes, thus leading to misleading conclusions. Furthermore, ectopic expression of Cre recombinase can lead to unwanted targeting [80,81,82]. In the context of DAT-Cre, we could recently show that several DAT-Cre mouse lines show ectopic expression of Cre recombinase in multiple brain areas that are not associated with monoaminergic neurotransmission. For example, certain amygdaloid subnuclei, septal nuclei and neurons in the lateral habenula, all of which contain VGLUT2 but not DAT, were strongly positive for the DAT-Cre transgenes [82]. Bearing this important observation in mind, it is possible that any results obtained using a DAT-Cre transgene to achieve gene-targeting of a floxed VGLUT2 or GLS1 allele, or any other floxed allele, might be dependent on recombination of the floxed gene in these non-monoaminergic areas. Possible “off-target” effects mediated by DAT-Cre are crucial to consider as they could have consequences on physiological and behavioral output, and hence on conclusions drawn from such experiments. Finding more selective tools to address DA–glutamate co-release in reward and addiction is highly relevant, not least in light of this new revelation.

### 2.8. DA–Glutamate Co-Release in Neuronal Plasticity within the Ventral Striatum

To address DA–glutamate co-release specifically in the adult mouse and to avoid any developmental effects upon VGLUT2-gene-targeting, we recently applied an inducible knockout approach [24]. In this study, we took advantage of a tamoxifen-inducible DAT-CreERT2 mouse line [83] in which DAT-Cre-expressing neurons will translocate Cre recombinase into the nucleus only upon tamoxifen administration. We could show that the spatial expression pattern of Cre recombinase in the inducible DAT-CreERT2 line is similar to the conventional DAT-Cre line implemented in our previous studies, while the temporal expression is regulated by the time at which the tamoxifen is provided. The new *Vglut2^f/f;DAT-CreERT2^* cKO mouse line was compared with our previously published *Vglut2^f/f;DAT-Cre^* cKO mouse line (described above) in a range of experiments in order to have the same VGLUT2 floxed allele targeted by the two different DAT-Cre-drivers. By comparative analysis of *Vglut2^f/f;DAT-CreERT2^* and *Vglut2^f/f;DAT-Cre^* cKO mice, we observed that when glutamate co-release was disrupted in adulthood, the mice displayed strikingly different behaviors in response to amphetamine and cocaine compared to mice with a developmentally-induced VGLUT2 targeting event. Unlike the drug-induced locomotor response observed in the *Vglut2^f/f;DAT-Cre^* cKO mice, which was lower than that shown by control mice, the inducible *Vglut2^f/f;DAT-CreERT2^* cKO mice displayed a similar level of drug-induced locomotor activity as controls [24]. Additionally, when electrophysiological experiments were performed to investigate how reduced levels of glutamate release from DA neurons might affect the plasticity observed after chronic drug administration, we found that inducible *Vglut2^f/f;DAT-CreERT2^* mice displayed characteristic increases in markers of synaptic plasticity already under baseline conditions. *Vglut2^f/f;DAT-CreERT2^* mice showed an increased AMPA/NMDA ratio on D1R-expressing MSNs compared to controls, and these changes were sufficient to occlude further increases normally seen following chronic cocaine administration [24] (Figure 8A). Increased AMPA/NMDA ratio is indicative of the presence of GluR2-lacking subunits of the AMPA receptor that have higher peak conductance and Ca^2+^ permeability and thus higher inward-rectifying properties (expressed as higher rectification index) [1]. While the inducible *Vglut2^f/f;DAT-CreERT2^* cKO mice showed an increased AMPA/NMDA ratio in baseline conditions, the rectification index of MSNs was unaltered (Figure 8B), reflecting mechanisms others than the replacement of GluR2 in AMPARs [24]. These mechanisms may include net increases in AMPA receptors, decreases in NMDA receptors, or both. Clearly, the mechanisms underlying the observed increases in baseline AMPA/NMDA ratios require further investigation to fully understand this type of regulation. However, already now, these data suggest that deletion of VGLUT2 in mature neurons induced changes in neuronal plasticity already under baseline conditions and that inducible *Vglut2^f/f;DAT-CreERT2^* cKO mice might have abnormal drug- or other stimuli-induced plasticity. More studies will be needed to find out how these observed changes might be of importance to addiction.

In summary, the main take-home message appearing from the various studies knocking out VGLUT2 in DAT-Cre neurons is that detectable neurocircuitry and behavioral changes do indeed occur upon targeted disruption of VGLUT2. No matter which combination of DAT-Cre transgene and floxed VGLUT2 allele that have been combined to probe DA–glutamate co-release, behavioral manifestations all point towards functional aspects of particular interest to reward and substance use/abuse. The availability of VGLUT2 in DA neurons of human individuals [49] might thereby be an important aspect to address in terms of vulnerability and pre-disposition to addiction, as briefly discussed further below.

### 2.9. Dopamine–Glutamate Co-Release: Implications for Reward Processing

How can all these observations from experimental animals be brought together into a comprehensive model to understand the potential role of DA–glutamate co-release in addiction? Despite the fact that additional studies are necessary in order to fully understand the role of co-release in behavioral regulation even under normal conditions, let alone under conditions of substance abuse, the findings reported in the research field so far can be summarized to form hypotheses related to the observed manifestations.

Drugs that are abused by human individuals can change brain physiology in many ways and eventually lead to characteristic and chronic behavioral patterns clinically known as substance use disorder. An abundance of studies has implicated several major neurotransmitter systems, including the DA and glutamate systems, in these phenomena while the role of combined DA and glutamate effects derived from the same neuron, i.e., DA–glutamate co-release, has only recently begun to be explored. Most studies in the addiction field have focused on the MSNs, as these neurons constitute the majority of neurons in the striatum. Similarly, initial studies of post-synaptic effects of DA–glutamate co-release have mainly been focused on recordings of MSN activity [24,56,57,63]. In the context of MSNs, Adrover and colleagues concluded that acute cocaine administration attenuates the DA neuron-induced EPSCs on MSNs through D2R activation based on the observation that the D2R antagonist sulpiride could reverse these effects [84]. Further, repeated cocaine administrations altered glutamatergic synapses between VTA terminals and NAc shell neurons in a withdrawal-dependent manner. After one day of withdrawal, there were no significant effects on the synapses, but when the VTA-NAc shell synapses were investigated three weeks after the last cocaine injection, a small but significant reduction in probability of neurotransmitter release was observed. This finding might suggest altered plasticity on the presynaptic level on DA–glutamate co-releasing populations, however, through the approach utilized, it cannot be excluded that the observed synapses were solely glutamatergic [85].

As described above, synaptic glutamate release in the striatum exerts different actions in a region- and neuron-type-dependent manner. DA–glutamate co-release has been demonstrated to not show the same distribution pattern throughout the striatum [57,86] and glutamate co-released into the NAc mSh affects ChIs more than MSNs or fast-spiking interneurons [58]. ChIs only constitute about 1–2% of striatal neurons in rodents, but they exert diffuse and profound effects on the physiology of the area [87,88]. ChIs display distinct burst-pause firing patterns and through ACh release, they can modulate presynaptic neurons by acting both at nicotinic and muscarinic ACh receptors [87,89]. Burst-pause firing rates in ChIs coincide with changes in the firing rate of DA neurons in response to reward-related, salient events but they convey different information compared to DA neuron firing [90]; pauses in their synchronous activity is thought to provide an optimal window where DA release due to increases in action potential frequency will have highest efficiency to promote the conveyed messages [11]. The psychostimulant amphetamine which increases DA release in terminal regions, has different effects depending on the region of the striatum and the neuronal type investigated. Thus, it attenuates the “burst-pause” neuronal activity observed on ChIs in the mSh in a dose-dependent manner, as shown by Chuhma et al. [58]. Only high doses affect post-synaptic activity on ChIs in the dorsal striatum while they have little effects on NAc core and on the EPSCs of MSNs in NAc mSh [58]. While the effects of chronic administration of drugs were not investigated in the study by Chuhma and colleagues, these observations indicate that psychostimulants can induce plasticity in different ways depending on anatomical region that in some instances can lead to behavioral abnormalities associated with chronic drug intake.

While it is known that SNc DA neurons project to the dorsal striatum, it has also been well established that VTA DA projections to striatal areas follow a medio-lateral topographical organization in the sense that DA neurons of the medial aspect of the VTA project to the NAc mSh and DA neurons located in the lateral aspect of the VTA project to NAc core [91]. DA neurons in the medial aspect of the VTA, which project mainly to the NAc mSh, are the ones that show the highest VGLUT2 expression levels, as described above. These histological findings have recently been confirmed using novel approaches implementing dual intersectional systems combining promoter activity of both TH and VGLUT2 to achieve selectivity for DA–glutamate co-releasing neurons [44,92]. Using these new approaches, previous results could be confirmed, firmly demonstrating that TH+/VGLUT2+ double-positive cell bodies are located primarily in the medial VTA. Further, identification of NAc mSh projections showed that this region receives the highest percentage of DA–glutamate co-release compared to other parts of the striatum [44,92]. As discussed above, the NAc mSh is a region highly implicated in reward-related behavior and drug addiction.

Based on novel findings that glutamate released by DA neurons mostly affect ChIs in this area, physiological and behavioral effects of DA-glutamate co-release will be discussed in this context. As outlined in a recent comprehensive review by Mingote and colleagues [44], glutamate released from DA neurons modulates the activity of ChIs, initially by increasing their firing rate [58]. This increase is correlated to the release of ACh which acts through nicotinic pre-synaptic receptors to affect DA terminals and further increases DA levels [87,89]. Mingote and colleagues suggest that these subsequent DA increases are necessary to induce behavioral switching, meaning that the organism will be able to engage in alternative behavioral patterns not associated to reward occurrence [44] (Figure 9). This hypothesis is supported by experimental studies on extinction and behavioral observations on *DAT^IREScre/+^::GLS1lox/+* mice. As discussed above, behavioral studies of these mice showed that they exhibit potentiated latent inhibition compared to control mice [64]. Under normal conditions, control mice would switch their behavior when exposed to conflicting contingencies, but reduced glutamate from DA neurons in the NAc mSh in *DAT^IREScre/+^::GLS1lox/+* mice induced a potentiation of the preserved response resulting in enhanced latent inhibition [44,64]. Data from our *Vglut2^f/f;DAT-Cre^* cKO mice also support this model: *Vglut2^f/f;DAT-Cre^* cKO mice will continue operant responding in response to cocaine [61] or optogenetic stimulation cues (Figure 7) during extinction. It is possible that reduction of glutamate release from DA–glutamate co-releasing neurons projecting to NAc mSh prevents disinhibition of downstream circuits which under normal conditions would promote alternative strategies and eventually lead to more efficient reward obtaining behaviors. This is demonstrated through persevered behaviors that can be related to cue-induced relapse in human addicts. For example, pre-clinical studies have demonstrated higher levels of VGLUT2 in terminals of alcohol-preferring rats after alcohol deprivation [93], and severe alcohol use disorder has been associated with polymorphisms in VGLUT2 [94]. While not making a distinction between VGLUT2 in glutamatergic neurons versus DA–glutamate co-releasing neurons in these studies, it is interesting and provoking to consider that alterations in VGLUT2 levels can affect both types of glutamatergic transmission in NAc mSh. Finally, it is possible that disturbed DA–glutamate co-release is responsible for drug-related behavioral manifestations due to overall altered DA tone and consequently to disturbances in behavioral switching.

### 2.10. Whole-Brain Analysis and Improved Selectivity in Animal Models Should Enhance Current Knowledge of DA-Glutamate Co-Releasing Neurons

While most of the published studies regarding DA–glutamate co-release have focused on the projections from the VTA to the NAc, it has also been reported that co-releasing neurons of the VTA project to a broader set of limbic regions in the rodent brain. The medial prefrontal cortex (mPFC), amygdala and hippocampus have all been shown to receive DA–glutamate co-releasing fibers from the VTA with different synaptic strengths [39,47,86,95,96,97]. As these brain areas have been strongly implicated in reward processing and drug-induced behavioral adaptations, future studies could aim to investigate the role of DA–glutamate co-release in each of these areas both under normal conditions and in models of disorders, such as addiction and schizophrenia. Furthermore, and consistent with the regional heterogeneity observed within the striatum, neurons of dual DA–glutamate profiles might have different effects on neurocircuitry depending on the target area. For instance, DA neurons projecting to the mPFC have been shown to code for aversive events [98,99] while DA release in the mPFC has the potential to alter dopaminergic and behavioral responses of subcortical areas to both natural stimuli [100] and addictive drugs [101]. For example, one question remaining to be addressed is how mesocortical DA–glutamate co-releasing neurons might influence subcortical responses to salient stimuli. In summary, by opening up for studies covering a brain perspective beyond the striatum, it seems likely that the knowledge of how DA–glutamate co-release affects physiology and behavior of relevance to both health and disease could be increased further.

Another point worth considering to forward current knowledge is the availability of animal models. It has been discussed in the literature how the lack of appropriate animal models has made it challenging to fully address behavioral roles of DA–glutamate co-release [19,22]. While this is indeed true, as outlined above, by implementing Cre-Lox-transgenics in rodents, the results of several different studies converge towards a strong implication for a role of DA–glutamate co-release in reward processing of relevance to addiction [54,61,62,63,64]. To now reach further, a higher level of selectivity is on the wish-list. For example, the recently described intersectional approaches ([44,92] and described above) should prove useful for further investigation of DA–glutamate co-release. To advance selectivity when creating new animal models, a first step forward might be to increase the level of anatomical and molecular knowledge of neurons that possess the ability for co-release.

By enhancing current knowledge of moleculary defined subpopulations in the VTA, it may be possible to increase the level of resolution in the anatomical-functional mapping of DA-glutamate co-releasing neurons. In this context, we have recently demonstrated that some, but not all, DA–glutamate co-releasing neurons in the VTA express the NeuroD6 gene, a finding which opens up for subtyping DA–glutamate co-releasing neurons based on molecular profiles beyond TH, DAT and VGLUT2 [102]. Since our identification of VGLUT2 expression in subsets of DA neurons from early embryonal development ([54] and discussed above), we have viewed the VGLUT2-positive DA neuronal population as a distinct subpopulation within the VTA, in accordance with current literature. Based on our initial study [54], we reasoned that the VTA might contain additional subpopulations that, if they could be identified by a molecular profile, could be used to dissect out the causality between distinct neuronal activity in the VTA and behavioral regulation of importance to reward and addiction. To search for additional gene expression patterns that, beyond VGLUT2, might distinguish DA neurons from each other, we performed a microarray analysis followed through with systematic histological validation. Through several steps of analyses, we could identify a number of gene expression patterns that were selective for neuronal groups within the VTA, suggesting that they might represent molecularly definable VTA subpopulations [23]. Similar types of studies have been performed in the pre-clinical DA field focused on Parkinson’s disease, which have substantially enriched molecular knowledge of both the VTA and the adjacently located SNc [103,104]. Recently, the advancement of transcriptomics analyses has enabled further gene expression analysis of the VTA and SNc [105,106]. All studies mapping out gene expression patterns in a particular brain location are of particular interest as they provide molecular tools that enable the anatomical-functional dissection of “subpopulations” (or “subgroups” or “subtypes”) of neurons.

In our microarray screen comparing gene expression in the VTA and the SNc, we identified the gene encoding the neurogenic basic helix-loop-helix transcription factor NeuroD6 as enriched in the VTA [23]. NeuroD6, which has also been reported by others [105,107,108], was found primarily in DA neurons located in the medial aspect of the VTA, suggesting that NeuroD6 expression represents a distinct subpopulation within the group of medially positioned VTA DA neurons that primarily project to the NAc Sh [23]. We also found that some, but not all, NeuroD6 VTA DA neurons were positive for VGLUT2 (Figure 10). This finding leads us to propose that the DA–glutamate co-releasing phenotype can be dissociated further based on molecular identity. While anatomically interesting, the most important aspect of molecular profiling is when it can be coupled to functional output. To directly test if we could identify a distinct role in behavioral regulation mediated by the newly discovered NeuroD6 VTA subpopulation, we implemented optogenetics using transgenic NeuroD6-Cre mice (also known as NEX-Cre) to direct optogenetic activation selectively to this NeuroD6-positive subpopulation. NeuroD6-Cre mice were compared with DAT-Cre and VGLUT2-Cre mice in a range of optogenetics-based analyses [102]. The experiments provided evidence for both glutamate and DA release in the NAcSh upon optogenetic stimulation in the VTA and could also demonstrate that selective stimulation of NeuroD6 VTA neurons led to significant place preference in a similar, but not identical, manner as when the entire VTA DA neuronal population was activated [102] (Figure 11). Activation of the NeuroD6 VTA subpopulation, with its mixed DA–glutamate co-releasing and non-co-releasing properties, is thereby sufficient to induce a distinct behavioral response [102]. This new finding should be well-worth exploring further in the context of substance use/abuse. This study is an example of how molecular knowledge of distinct VTA neurons can be used to implement available animal models in a new way or to even create new animal models with spatial and temporal selectivity for distinct neuronal subgroups at a level required to advance current knowledge of co-releasing neurons. By establishing direct causality between distinct subgroups of DA–glutamate co-releasing neurons and behavioral regulation of importance to reward and addiction, future studies can use this kind of knowledge to understand how regulation of co-release could be implemented clinically for the benefit of future interventive strategies in substance use disorders.

### 2.11. Concluding Remarks

This review has summarized the main findings derived from behavioral analyses in genetically modified mice produced with the aim of pin-pointing the putative relevance of DA–glutamate co-release in behavioral regulation. While different kinds of transgenic mouse lines have been generated and analyzed in various methodological paradigms spanning from baseline locomotion to self-administration of abusive substances, all studies share the common conclusion that behaviors of particular interest for addiction are altered when glutamate co-release is disrupted (results summarized in Table 1 and illustrated in Figure 12). Many questions remain to be answered, but so far, the experimental findings point towards glutamate co-release as a putative target in the treatment of substance use disorder, including prevention of relapse. The study of DA–glutamate co-release for clinical purposes would benefit from a higher degree of selectivity in experimental approaches which could aid in determining how this type of signaling could be used as a tool in prevention and treatment of addiction.

## Figures and Tables

**Figure 1 jcm-08-01887-f001:**
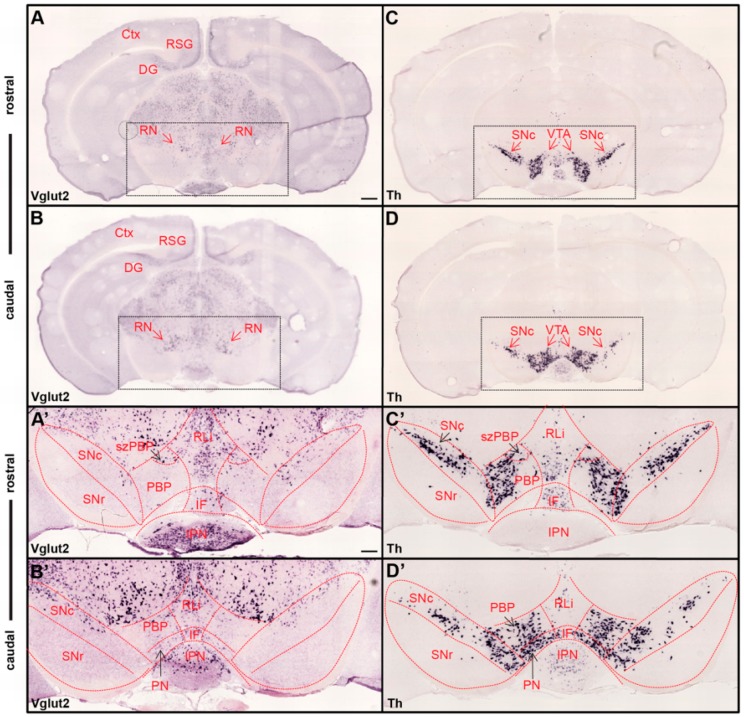
Ample Vglut2 mRNA-positive cells throughout dorsal and ventral midbrain with more sparse expression within the dopaminergic area. Colorimetric in-situ hybridization showing overview of Vglut2 (**A**,**B**) and Th (**C**,**D**) mRNA in midbrain sections of wildtype adult mouse at two rostro-caudal levels. (**A**,**B**) Vglut2 mRNA is abundant throughout the midbrain with strong signals in e.g., the red nucleus (RN), retrosplenial group of the cortex (RSG) and dentate gyrus (DG), and weaker signals in the ventral tegmental area (VTA) and substantia nigra *pars compacta* (SNc). (**C**,**D**) Th mRNA is selectively localized in dopaminergic neurons of the VTA and SNc; its mRNA signal is implemented to visualize these areas. Dotted square around the VTA and SNc (scale bar 500 mm) presented as closeups in (**A’**–**D’**; scale bar 200 mm). (**C’**,**D’**) SNc, substantia nigra *pars reticulata* (SNr) and subregions of VTA outlined in Th closeups and superimposed on Vglut2 closeups (**A’**,**B’**). (**C’**,**D’**) Th mRNA was strongly localized in the SNc and within the parabrachial pigmented area (PBP) and paranigral nucleus (PN) of the VTA with a weaker signal in the rostral linear nucles (RLi) and caudal aspect of the interfascicular nucleus (IF). (**A’**,**B’**) Within the VTA, Vglut2 mRNA was detected in the PBP, PN, RLi, and IF as well as within the medially-located subzone of the PBP (szPBP) while no Vglut2 mRNA was detected in the GABAergic SNr area. Additional abbreviations: Ctx, Cortex; IPN, interpeducular nucleus. Reprinted from Papathanou et al., 2018 [24].

**Figure 2 jcm-08-01887-f002:**
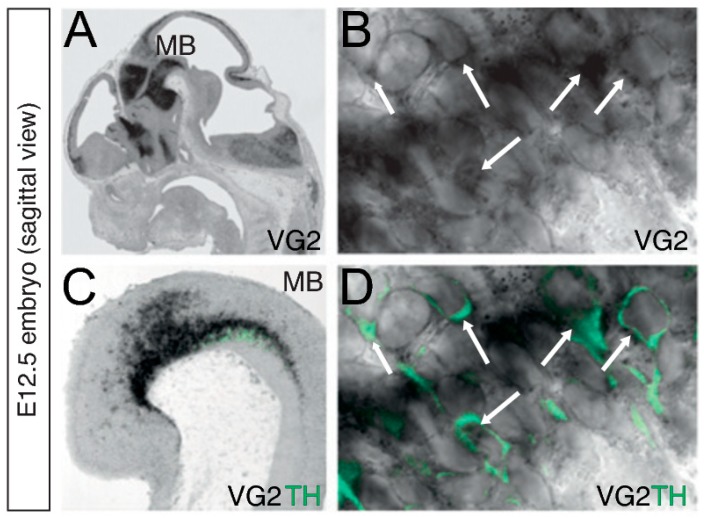
Vglut2 mRNA and Tyrosine hydroxylase (TH) immunoreactivity in midbrain DA neurons. (**A**–**D**) In-situ hybridization for Vglut2 (VG2) mRNA (black) combined with fluorescent immunohistochemistry for TH (green) on sagittal sections of an E12.5 embryo. Vglut2 mRNA is detected in multiple regions in the embryo, including the ventral midbrain (MB) where DA neurons develop (**A**,**B**). Vglut2 mRNA is co-localized with TH-immunoreactivity in DA neurons within the MB (**C**,**D**). Arrows indicate Vglut2-positive cytoplasm (**B**) and Vglut2/TH double-positive neurons (**D**) (Magnification: B,D 300x). Reprinted from Birgner et al., 2010 [54].

**Figure 3 jcm-08-01887-f003:**
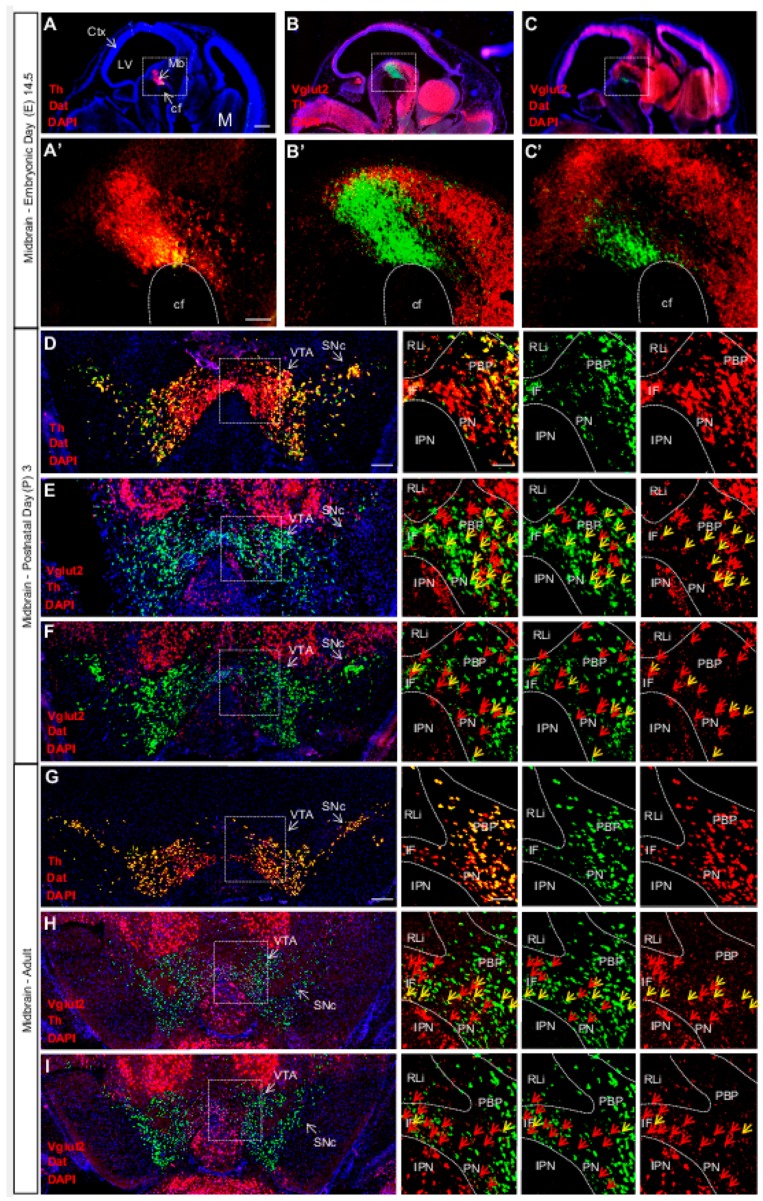
Vglut2, Th and Dat mRNA co-localization within certain VTA dopamine (DA) neurons is sparse at E14.5, peaks around birth and is subsequently down-regulated in adulthood. Double fluorescent in-situ hybridization for Th (red), Dat (green) and Vglut2 (red) mRNA, respectively, on wildtype mouse midbrain sections. (**A**–**C**) Sagittal sections of E14.5 embryo. Dotted square around the area of developing midbrain DA neurons (**A**–**C**) with close-ups in (**A’**–**C’**). (**A**) Th and Dat mRNA show co-localization (yellow) in the ventral midbrain (scale bar 500 mm). (**A’**) higher magnification of insets (scale bar 100 mm); (**B**,**B’**) Th and Vglut2 mRNA expression in the midbrain. (**C**,**C’**) Dat and Vglut2 mRNA show sparse detection in the midbrain. (**D**–**F**) Coronal sections of ventral midbrain in pups of postnatal day (P) 3. (**D**) Th and Dat show ample co-localization (yellow) in the lateral VTA and SNc (scale bar 250 mm, inset 100 mm). (**E**) Th and Vglut2 mRNA and (**F**) Dat and Vglut2 mRNA prominently co-localize (yellow) at this age in the IF, PBP and PN areas (arrows) but not in the RLi of the VTA. (**G**–**I**) Coronal sections of the adult midbrain (10 weeks; scale bar 250 mm, inset 100 mm). (**G**) Th and Dat mRNA co-localization (yellow) remains strong; whilst the level of co-localization between (**H**) Th and Vglut2 and (**I**) Dat and Vglut2 mRNAs is lower than at P3 (arrows). Yellow arrows show co-localization green (Dat) and red (Vglut2) channel, red arrows show red (Vglut2) channel (Postnatal Day (P) 3 *n* = 3; adult *n* = 3). Abbreviations: cf, cephalic flexture; Ctx, cortex; IF, interfascicular nucleus; IPN, interpeducular nucleus; LV, lateral ventricle; M, medulla; Mb, midbrain; PBP, parabrachial pigmented area; PN, paranigral nuclei; RLi rostral linear nucleus; SNc, Substantia nigra pars compacta; VTA, Ventral tegmental area. Reprinted from Papathanou et al., 2018 [24].

**Figure 4 jcm-08-01887-f004:**
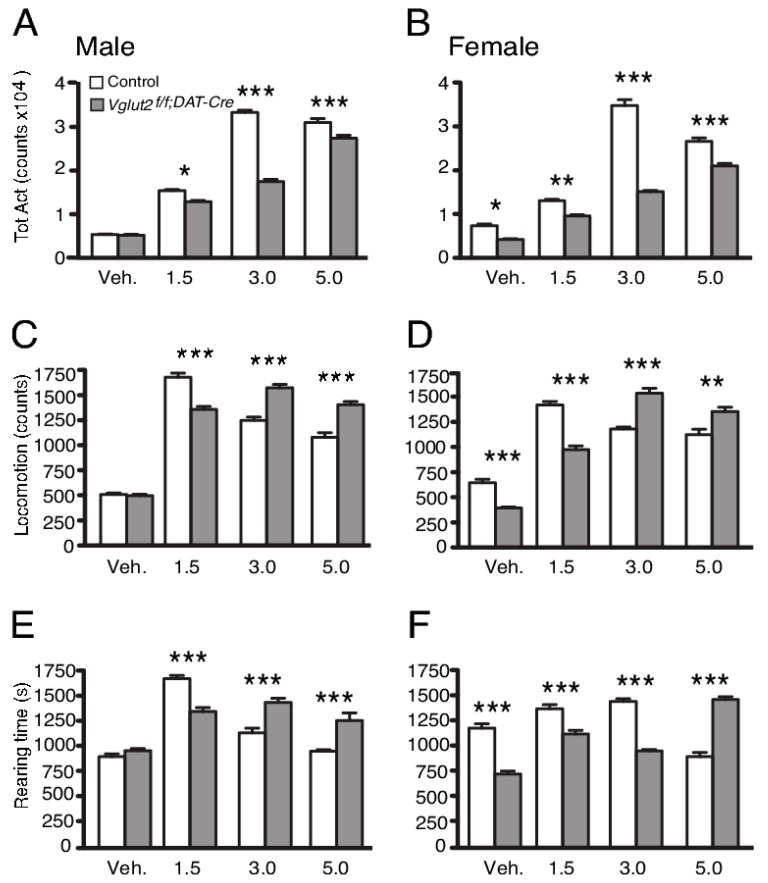
*Vglut2^f/f;DAT-Cre^* mice show blunted behavioral response to amphetamine, as measured in a home-cage environment. (**A** and **B**) Male and female knockout mice showed a reduced behavioral response to amphetamine as shown by significantly lower total activity at all three doses of amphetamine. (**C** and **D**) Male and female knockout mice display a shift in dose-response to amphetamine-induced locomotion. (**E** and **F**) Male and female knockout mice show a different dose-response profile regarding rearing behavior in response to amphetamine. A right shift in dose-response in locomotion is apparent in the *Vglut2^f/f;DAT-Cre^* mice. Data were analyzed with one-way ANOVA followed by Tukey’s post-hoc test when appropriate. Data are presented as mean ± SEM (*n* = 9). *, *P* < 0.05; **, *P* < 0.01; ***, *P* < 0.001. Reprinted from Birgner et al., 2010 [54].

**Figure 5 jcm-08-01887-f005:**
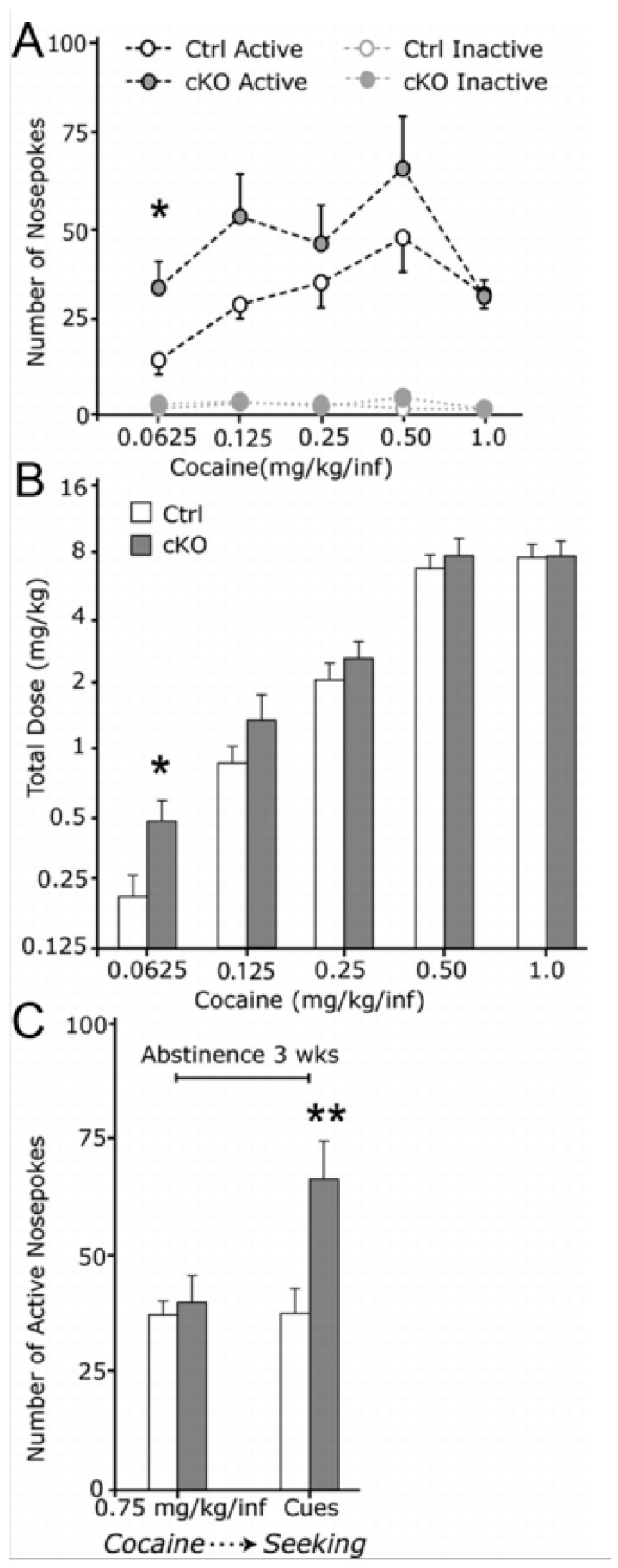
Elevated operant responding for low-dose cocaine and drug-paired cues during extinction in cKO mice. (**A**) Nosepoke response of food-trained mice implanted with indwelling intravenous catheters and allowed to nosepoke for cocaine infusions (controls, *n* = 7; cKO, *n* = 7). (**B**) Total dose self-administered at the different cocaine concentrations in (**A**). (**C**) Cocaine seeking, i.e., responding for light and sound cues previously associated with cocaine in the absence of the drug, of mice from A that had been responding for 0.75 mg/kg per infusion (inf) and subsequently subjected to 21 d of forced cocaine abstinence (controls, *n* = 6; cKO, *n* = 5). Group data represent mean ± SEM. **p* < 0.05; ***p* < 0.01 versus Ctrl. Reprinted from Alsiö et al., 2011 [61].

**Figure 6 jcm-08-01887-f006:**
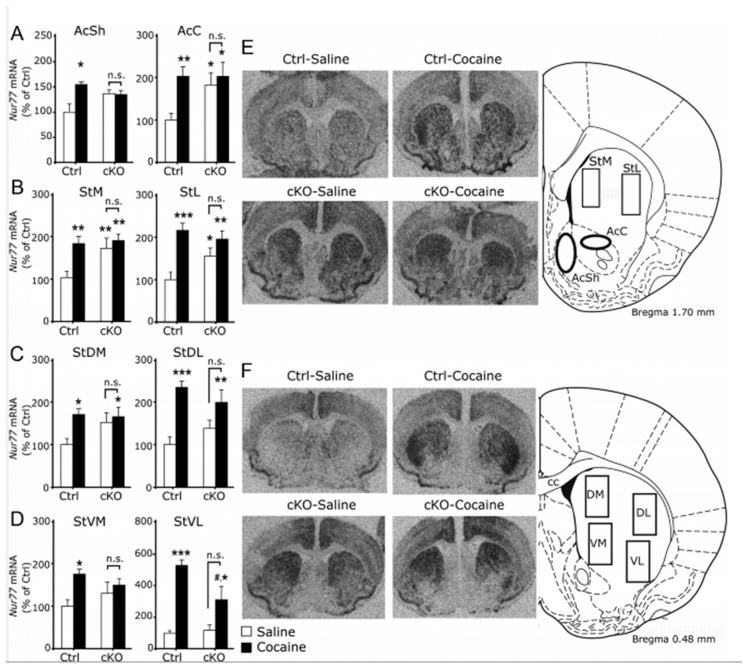
High expression of Nur77 under baseline conditions in cKO mice. (**A**–**D**) Quantitative in-situ hybridization of Nur77 mRNA levels in Ctrl and cKO mice treated with saline (Ctrl, *n* = 6; cKO, *n* = 7) or cocaine (Ctrl, *n* = 5; cKO, *n* = 4). (**A**) Levels of Nur77mRNAin AcSh and AcC (bregma, 1.70 mm). (**B**) Nur77mRNAlevels in medial (StM) and lateral (StL) rostral striatum (bregma, 1.70 mm). (**C**) Nur77 transcript levels in the caudodorsal striatum (bregma, 0.48 mm). (**D**) Nur77 levels in the caudoventral striatum (bregma, 0.48). (**E**,**F**) Representative autoradiograms of Nur77 mRNA in-situ hybridization signals in the rostral striatal area (**E**; bregma, 1.70 mm) and in the caudal striatal area (**F**; bregma, 0.48 mm) with schematic illustrations to the right showing the location of the analyzed regions. Group data represent mean ± SEM expressed as percentage of Ctrl (saline). **p* < 0.05, ***p* < 0.01, and ****p* < 0.001 versus Ctrl saline group; #*p* < 0.05 versus Ctrl cocaine group. cc, Corpus callosum. Reprinted from Alsiö et al., 2011 [61].

**Figure 7 jcm-08-01887-f007:**
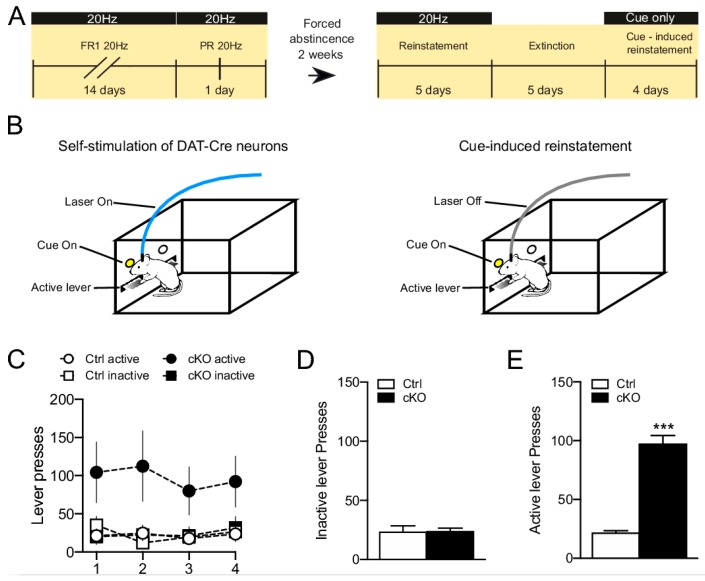
(**A**). Timeline of the intracranial optogenetic self-stimulation (ICSS) experiments. (**B**). Schematic representation of the ICSS procedure and the cue-induced reinstatement phase. (**C**). Active and inactive lever presses during the cue-induced reinstatement period for cKO and control mice. (**D**,**E**). Average of inactive (**D**) and active (**E**) during the cue-induced reinstatement phase cKO and Ctrl mice. Data are presented as mean ± SEM. *** *p* < 0.001 vs. ctrl. Ctrl *n* = 3, cKO *n* = 3. Original data, Bimpisidis and Wallén-Mackenzie, pilot study, 2019.

**Figure 8 jcm-08-01887-f008:**
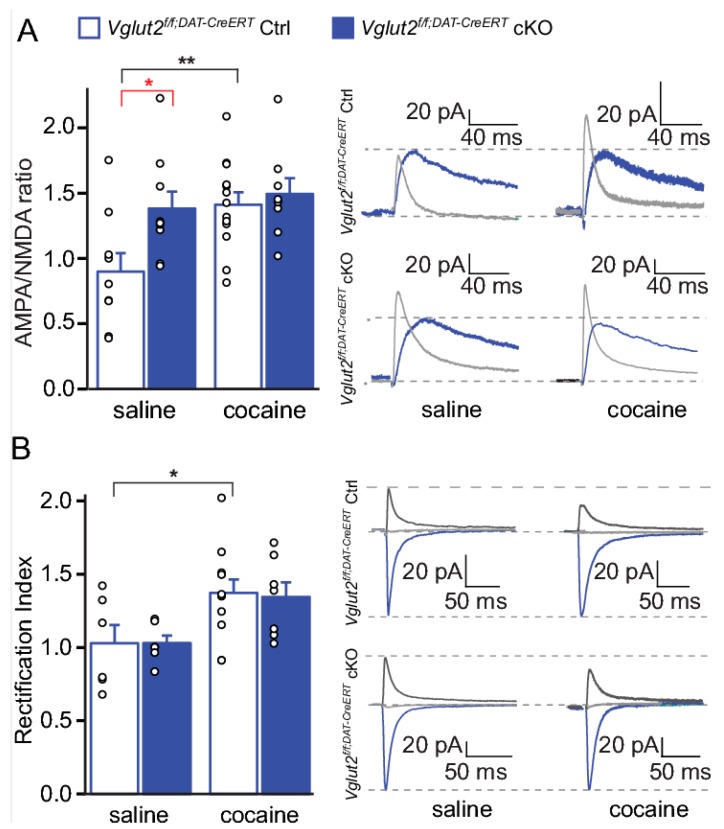
VGLUT2 targeting in mature DA neurons results in an elevated baseline AMPA/NMDA ratio. (**A**) AMPA/NMDA ratio and raw traces of cells from *Vglut2^f/fDAT-CreERT^* cKO and ctrl mice treated with saline or cocaine for NMDA current (blue); AMPA current (light gray). (**B**) Rectification index (RI) and raw traces recorded cells from *Vglut2^f/fDAT-CreERT^* cKO and ctrl mice treated with saline or cocaine. eKO-DRD1 and eCtrl-DRD1 mice treated with saline or cocaine at −70 mV (blue), 0 mV (light gray) and +40 mV (dark gray). Two-way ANOVA Sidak post hoc (ANOVA ###*p* < 0.001; post hoc between genotype **p* < 0.05 and post hoc between treatment of same genotype: **p* < 0.05, ***p* < 0.01, ****p* < 0.001; RI saline: saline *Vglut2^f/fDAT-CreERT^* Ctrl, *n* = 6, *Vglut2^f/fDAT-CreERT^* cKO, *n* = 6; cocaine: *Vglut2^f/fDAT-CreERT^* Ctrl, *n* = 12; *Vglut2^f/fDAT-CreERT^* cKO, *n* = 7; AMPA/NMDA ratio saline:: *Vglut2^f/fDAT-CreERT^* Ctrl, *n* = 9, *Vglut2^f/fDAT-CreERT^* cKO, *n* = 9; cocaine: *Vglut2^f/fDAT-CreERT^* Ctrl, *n* = 13, *Vglut2^f/fDAT-CreERT^* cKO, *n* = 7). Whole-cell patch clamp experiments performed on slices 10 days after last saline or cocaine injection. Reprinted from Papathanou et al., 2018 [24].

**Figure 9 jcm-08-01887-f009:**
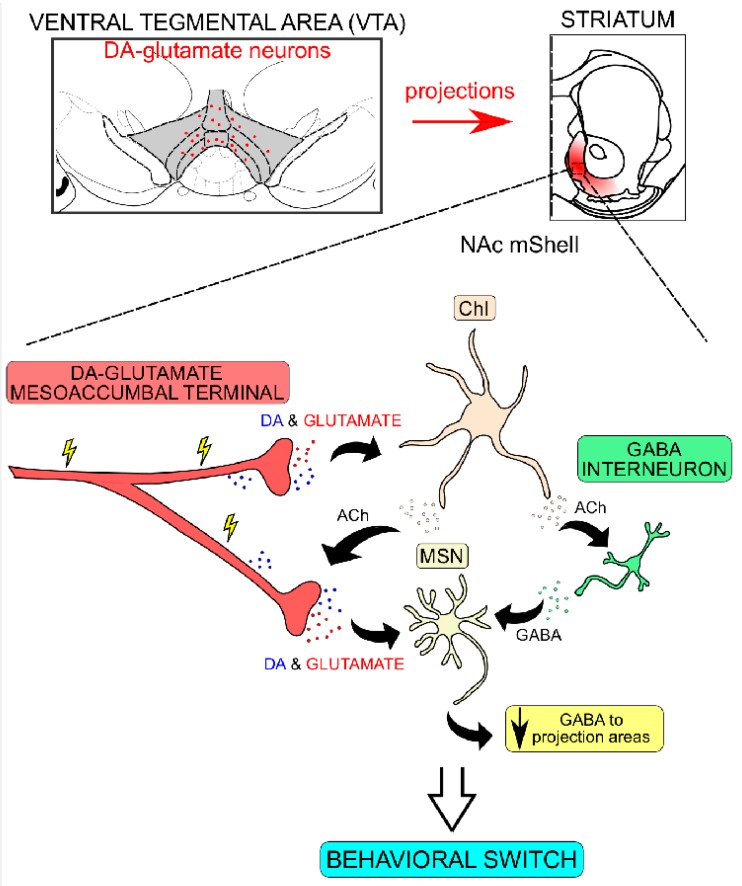
Simplified schematic model of down-stream neurocircuitry effects upon dopamine (DA)–glutamate co-release in the ventral striatal area leading to behavioral output. DA–glutamate co-releasing neurons (red) are located primarily in the medial part of the ventral tegmental area (VTA; gray) from where they project to the nucleus accumbens medial shell (NAc mShell). Burst-firing of these neurons leads to a release of DA and glutamate from mesoaccumbal nerve terminals which subsequently act on DA and glutamate receptors located on cholinergic interneurons (ChIs) and medium spiny neurons (MSNs) in the NAc mShell. DA and glutamate neurotransmission via receptors located on ChIs leads to synchronized activity and acetylcholine (ACh) release. ACh subsequently acts on ACh receptors located in VTA presynaptic terminals to further increase neurotransmitter release. DA and glutamate release, together with increased release of GABA from GABAergic interneurons, also leads to inhibition of the GABAergic MSNs, which in turn leads to disinhibiton of target areas and allows for the occurence of behaviors associated with alternative strategies to obtain reward, a “behavioral switch”. Drawing based on original illustration by Mingote et al., 2019 [44].

**Figure 10 jcm-08-01887-f010:**
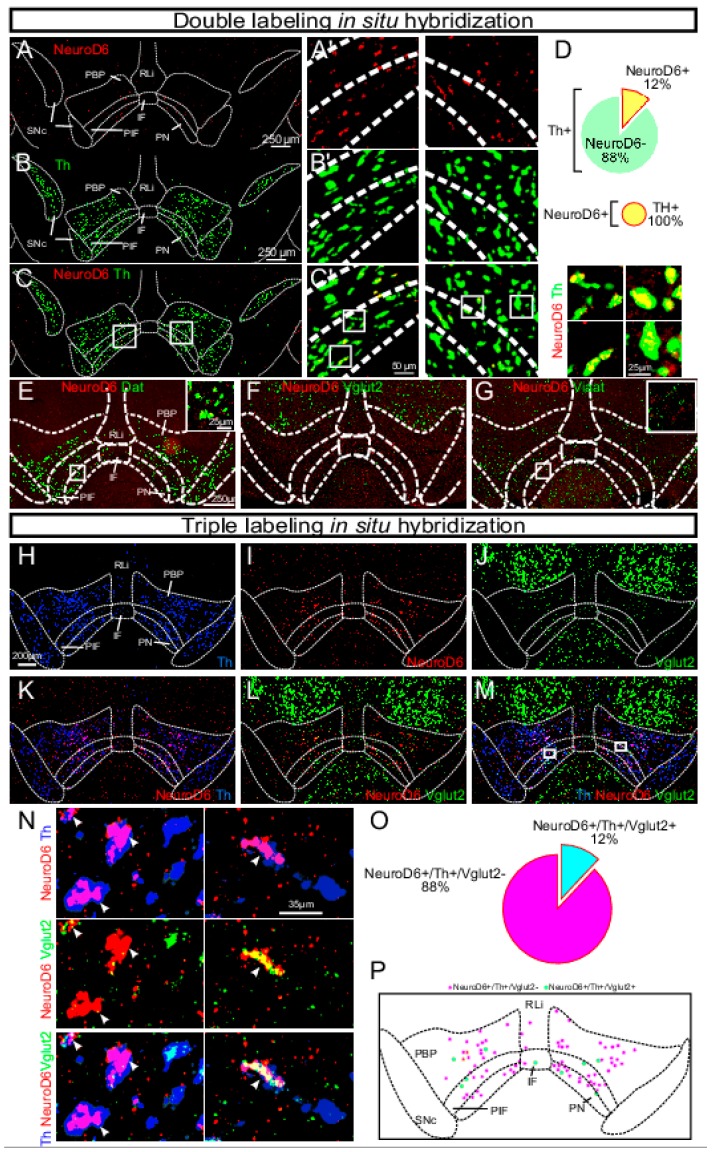
NeuroD6 mRNA is found in a modest population of the VTA and co-localizes with dopaminergic markers and partially with a glutamatergic marker. (**A**–**G**) Double FISH in the ventral midbrain of adult wild-type mice detecting the following mRNAs. **A**, **A’**, NeuroD6 (red). (**B**,**B’**) Th (green). (**C**,**C’**) NeuroD6 (red) and Th (green). Th/NeuroD6 mRNA that overlap are shown in yellow. Low magnification to the left; close-ups to the right. Schematic outline shows borders for SNc and subregions of VTA: PN, PIF, PBP, IF, RLi. (**D**) Quantification of percentage of NeuroD6-positive cells among all Th VTA cells; all NeuroD6 cells are positive for Th mRNA. (**E**) NeuroD6 (red) and Dat (green), inset with high magnification of Dat/NeuroD6 mRNA overlap (yellow). (**F**) NeuroD6 (red) and Vglut2 (green). (**G**) NeuroD6 (red) and Viaat (green), inset with high magnification of Viaat-negative/NeuroD6-positive (red). (**H**–**P**) Triple-labeling FISH in the ventral midbrain of adult wild-type mice detecting: (**H**) Th (blue); (**I**) NeuroD6 (red); (**J**) Vglut2 (green) mRNAs and their co-localization: (**K**) NeuroD6/Th; (**L**) NeuroD6/Vglut2; (**M**) Th/NeuroD6/Vglut2. Cellular closeups: (**N**) NeuroD6/Th (top), NeuroD6/Vglut2 (middle), and Th/NeuroD6/Vglut2 (bottom). Arrows point to NeuroD6 mRNA-positive cells. (**O**) Quantification of percentage of NeuroD6+/Th+/Vglut2+ and NeuroD6+/Th+/Vglut2- neurons of the VTA. (**P**) Schematic illustration of distribution pattern of NeuroD6+/Th+/Vglut2+ and NeuroD6+/Th+/Vglut2- neurons within the VTA (same as shown with experimental data in **M**). NeuroD6+/Th+/Vglut2- cells in magenta; NeuroD6+/Th+/Vglut2+ cells in cyan. VTA, ventral tegmental area; SNc, substantia nigra pars compacta; PBP, parabrachial pigmented nucleus; PN, paranigral nucleus; PIF, parainterfascicular nucleus; RLi, rostral linear nucleus; IF, interfascicular nucleus. FISH, fluorescent in situ; Dat, Dopamine transporter; Th, Tyrosine hydroxylase; Vglut2, Vesicular glutamate transporter 2; Viaat, Vesicular inhibitory amino acid transporter. Reprinted from Bimpisidis et al., 2019 [102].

**Figure 11 jcm-08-01887-f011:**
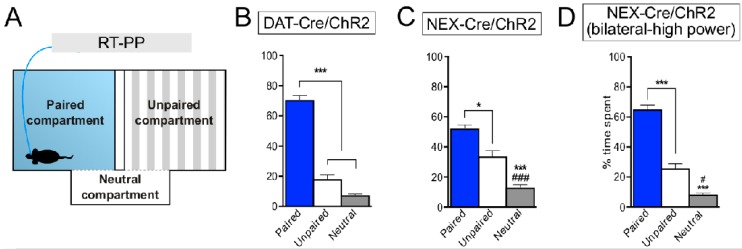
Optogenetic activation of NeuroD6 VTA neurons induces place preference. (**A**) Schematic drawing of the real-time place preference (RT-PP) experimental setup. (**B**–**D**) average percentage of time spent in each compartment during 4 days of RT-PP ± SEM (bar graphs; right; **p* < 0.05, ****p* < 0.001 vs. light-paired compartment; #*p* < 0.05, ##*p* < 0.01, ###*p* < 0.001 vs. unpaired compartment). DAT-Cre, *n* = 10; NEX-Cre, *n* = 5, high-power stimulation of bilaterally-injected NEX-Cre mice, *n* = 4. Reprinted from Bimpisidis et al., 2019 [102].

**Figure 12 jcm-08-01887-f012:**
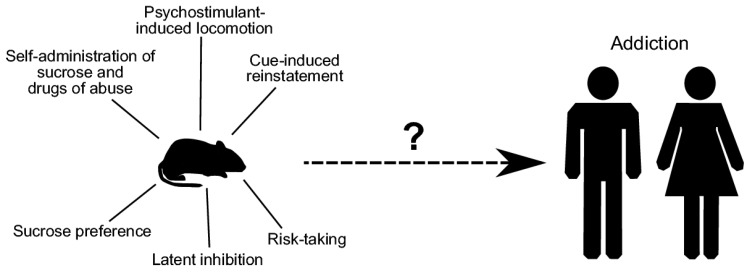
Schematic illustration of behaviors relevant to addiction confirmed in experimental mice to be altered upon disruption of dopamine–glutamate co-release (see text and Table 1). Further studies will be needed to outline how these behaviors relate to typical behaviors of addiction displayed by humans and how dopamine–glutamate co-release can be used as a future target for prevention and treatment.

**Table 1 jcm-08-01887-t001:** List of publications in which dopamine-glutamate co-release has been targeted to probe its putative behavioral roles. Main results summarized. – indicates no change; ↑ increase; ↓ decrease. Abbreviations: RT-PP, real-time place preference; VTA, ventral tegmental area.

Study (Listed in Alphabetical Order)	Strategy to Target DA-Glutamate Co-Release	Test	Effect
**Alsiö et al., 2011 [61]**	*Vglut2^f/f;DAT-Cre^*	Sucrose preference	lower threshold
		Sucrose self-administration	↑ responding under food restriction
			↑ consumption when fed *ad libitum*
		Cocaine self-administration	↑ for lower doses
			↑ responding for cocaine-associated cues
**Birgner et al., 2010 [54]**	*Vglut2^f/f;DAT-Cre^*	Rotarod (crude motor coordination)	−
		Beam walking (fine motor coordination)	−
		Elevated plus maze (anxiety)	↑ latency to move
		Multi-variate concentric square field (anxiety and risk analysis)	↑ risk-taking behavior
		Forced swim test (depression)	−
		Radial maze	−
		Acute amphetamine	dose-dependent alterations
**Fortin et al., 2012 [62]**	*Vglut2^f/f;DAT-Cre^*	Rotarod (crude motor coordination)	↓ motor performance
		Forced swim test (depression)	↓ latency to immobility
		Spontaneous activity in novel environment	↓ horizontal activity
		Acute amphetamine	↓ locomotor responses
		Acute cocaine	↓ locomotor responses
**Hnasko et al., 2010 [63]**	*Vglut2^f/f;DAT-Cre^*	Spontaneous locomotion	−
		Rotarod (crude motor coordination)	−
		Acute cocaine	↓ locomotor responses
		Cocaine sensitization	−
		Cocaine conditioned place preference (CPP)	−
**Mingote et al., 2017 [64]**	*DAT^IREScre/+^::GLS1lox/+*	Rotarod (crude motor coordination)	−
		Novelty induced locomotion	−
		Elevated plus maze (anxiety)	−
		Fear conditioning	−
		Acute amphetamine	−
		Amphetamine sensitization	↓
		Latent inhibition	↑
**Papathanou et al., 2018 [24]**	*Vglut2^f/f;DAT-CreERT2^*	Amphetamine and cocaine sensitization	− (baseline AMPA/NMDA ratio altered)
**Wang et al., 2017 [65]**	*Vglut2^f/f;DAT-Cre^*	RT-PP (optogenetics in VTA)	−
		Self-stimulation (optogenetics in VTA)	no effects on acquisition ↓ responses in higher laser power stimulation

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
