# Peer review of "Neurocircuitry of Reward and Addiction: Potential Impact of Dopamine–Glutamate Co-release as Future Target in Substance Use Disorder"

_jcm, 2019, doi:10.3390/jcm8111887_

Round 1

Reviewer 1 Report

Bimpisidis and Wallén-Mackenzie review neural underpinnings of reward and addiction focusing on dopamine neuron glutamate cotransmission. The review is timely and comprehensive, distilling a growing synthesis regarding the role of dopamine neuron glutamate cotransmission in substance use/dependence.

Comments

Lines 2-4, Title: “co-release” is used, but “cotransmission” would be preferable. Cotransmission is the release of more than one neurotransmitter from a single neuron. Co-release is release of more than one neurotransmitter from a synaptic vesicle. See for instance, Vaaga et al. Curr Opin Neurobiol, 2014 (page 25), “Co-transmission, the ability of a neuron to release multiple transmitters, has long been recognized in selected circuits. However, the release of multiple primary neurotransmitters from a single neuron is only beginning to be appreciated. Here we consider recent examples of co-transmission as well as corelease — the packaging of multiple neurotransmitters into a single vesicle.” Given now the predominant evidence that dopamine and glutamate are released at separate release sites (Zhang et al. 2015; Silm et al. 2019), and not by co-release, use of “cotransmission” would be more accurate.

Lines 141-145: There appears to be a contradiction: referencing Fig. 2, authors state, “we could readily identify VGLUT2-TH co-localization already at E12.5 [50]”, but then referencing Fig. 3, “Almost no co-localization between VGLUT2 with either TH or DAT was detected at E14.5”. Do the authors intend to state that there is colocalization at E12.5, none at E14.5, and then “substantial co-localization … in the newborn mouse”?

Lines 158-164: Panels A-D are from Birgner et al., 2010, Figure 1. Caption is copied, with some modifications, and refers to panels not included in the present ms. (e.g. DAT-cre activity). Use of panel letters is inconsistent. There are font problems with the magnifications. In the figure, the colorization of the panel letters has shifted. The relationship between panel A and C is not indicated; presumably C is an expanded view of A, but MB doesn’t appear to indicate the same structure in the two panels. B and D are the same field, adding TH immunostaining in D, so it would be better if they were shown in one panel, e.g. B1 and B2. “Control” in label on left is not relevant given that panels E-M of the original figure are not included. Although caption states that figure is modified from [50], no modification has been made, other than including only panels A-D, so copyright permission should be obtained.

Lines 209-212: This short prefatory paragraph should be incorporated into the next, as it is stylistically aberrant.

Lines 240-244: Unpublished results are mentioned regarding perinatal mortality of VGLUT2-Cre;VMAT2 lox/lox mice. The interpretation, “due to a breathing-phenotype similar as we observed when deleting VGLUT2 in all cells” does not make sense since it is VMAT2 and not VGLUT2 that is deleted here. Given the prevalence of VGLUT2 expression in monoaminergic neurons during gestation, it is more likely that the pups die due to failure to feed, or have autonomic irregularities, not breathing problems.

Line 254-255: “the psychostimulant amphetamine, which acts via the mesostriatal DA system.” Would be written better as “acts on/targetsthe mesostriatal DA system.”

Line 416: Up till now, the term has been dopamine-glutamate co-release, now it changes to glutamate-dopamine co-release – why?

Line 433: “conditioned” should be “conditional”

Line 437: “disturbed” does not communicate that sensitization is attenuated in GLS1(f/f;DAT-cre) mice.

Line 536: “Availability of VGLUT2 in DA neurons of human individuals” -- see Root et al., Scientific Reports, 2016.

Line 565: The statement that “ChIs make up only about 5% of the neurons in the striatum” exaggerated the number in rodents, which is 1- 2%.

Lines 620 and 707: The section title “Future perspectives” is premature, since further data are discussed. It would be better to revise the title accordingly, and then add a “Conclusions” title for the last paragraph.

Most of the figures are reprinted from previous publications of the authors’, but there are no statements regarding copyright permission having been obtained.

Some of the figures, e.g. Figure 3, are not reproduced at adequate resolution.

Table 1: “x” is used for “no change”, but a symbol with less positive valence would be better, e.g. a dash or a horizontal double arrow.

Author Response

Comments and Suggestions for Authors

Bimpisidis and Wallén-Mackenzie review neural underpinnings of reward and addiction focusing on dopamine neuron glutamate cotransmission. The review is timely and comprehensive, distilling a growing synthesis regarding the role of dopamine neuron glutamate cotransmission in substance use/dependence.

Answer: We thank you for this summary and address your comments below.

Comments

Lines 2-4, Title: “co-release” is used, but “cotransmission” would be preferable. Cotransmission is the release of more than one neurotransmitter from a single neuron. Co-release is release of more than one neurotransmitter from a synaptic vesicle. See for instance, Vaaga et al. Curr Opin Neurobiol, 2014 (page 25), “Co-transmission, the ability of a neuron to release multiple transmitters, has long been recognized in selected circuits. However, the release of multiple primary neurotransmitters from a single neuron is only beginning to be appreciated. Here we consider recent examples of co-transmission as well as corelease — the packaging of multiple neurotransmitters into a single vesicle.” Given now the predominant evidence that dopamine and glutamate are released at separate release sites (Zhang et al. 2015; Silm et al. 2019), and not by co-release, use of “cotransmission” would be more accurate.

Answer: Thank you for this important remark. We have used the concept “co-release” in our article to broadly define neurons that release both glutamate and dopamine, irrespective of mechanism. This terminology is based on current data and terminology of the field. Even articles produced in 2019 primarily use co-release over co-transmission when discussing neurons with “dual” identity (e.g. The Ventral Tegmental Area has calbindin neurons with the capability to co-release glutamate and dopamine into the nucleus accumbens. Mongia S, Yamaguchi T, Liu B, Zhang S, Wang H, Morales M. Eur J Neurosci. 2019 Jun 19. doi: 10.1111/ejn.14493.) and we have kept “co-release” to align with the field.

However, the importance of nomenclature has been considered in the revised manuscript. To clarify and define nomenclature and also to address how release of two neurotransmitters might take place in the context of synaptic vesicles, a new paragraph has been added to the manuscript, under the headline Vesicular glutamate transporters and the concept of DA-glutamate co-release” with citations of relevant references. A definition of “co-transmission” derived from El Mestikawy et al, 2011, has been included in this section which also includes a reference to Zhang et al, 2015. Please note that we also refere to other reviews for detailes concerning mechanisms of release.

Lines 141-145: There appears to be a contradiction: referencing Fig. 2, authors state, “we could readily identify VGLUT2-TH co-localization already at E12.5 [50]”, but then referencing Fig. 3, “Almost no co-localization between VGLUT2 with either TH or DAT was detected at E14.5”. Do the authors intend to state that there is colocalization at E12.5, none at E14.5, and then “substantial co-localization … in the newborn mouse”?

Answer: Indeed, this is the case. Based on our previously published data, our studies have shown that Vglut2 is more abundant in midbrain dopamine neurons at E12.5 than E14.5 and then goes up again around birth (Birgner et al, PNAS, 2010, Papathanou, eNeuro 2018). In a current manuscript in revision (Dumas & Wallén-Mackenzie, 2019), we have addressed the early expression patterns of Vglut2 in more detail (E9.5-E14.5) and can confirm this finding yet again. To clarify the point of temporal regulation of VGLUT2 during embryogenesis, this section in the manuscript has been updated with new text and we have also included a short section about these new data to give a more complete view of the embryonal expression of VGLUT2, now that we have this data available.

Lines 158-164: Panels A-D are from Birgner et al., 2010, Figure 1. Caption is copied, with some modifications, and refers to panels not included in the present ms. (e.g. DAT-cre activity). Use of panel letters is inconsistent. There are font problems with the magnifications. In the figure, the colorization of the panel letters has shifted. The relationship between panel A and C is not indicated; presumably C is an expanded view of A, but MB doesn’t appear to indicate the same structure in the two panels. B and D are the same field, adding TH immunostaining in D, so it would be better if they were shown in one panel, e.g. B1 and B2. “Control” in label on left is not relevant given that panels E-M of the original figure are not included. Although caption states that figure is modified from [50], no modification has been made, other than including only panels A-D, so copyright permission should be obtained.

Answer: Corrected.

Lines 209-212: This short prefatory paragraph should be incorporated into the next, as it is stylistically aberrant.

 Answer: Corrected.

Lines 240-244: Unpublished results are mentioned regarding perinatal mortality of VGLUT2-Cre;VMAT2 lox/lox mice. The interpretation, “due to a breathing-phenotype similar as we observed when deleting VGLUT2 in all cells” does not make sense since it is VMAT2 and not VGLUT2 that is deleted here. Given the prevalence of VGLUT2 expression in monoaminergic neurons during gestation, it is more likely that the pups die due to failure to feed, or have autonomic irregularities, not breathing problems.

Answer: We agree that this paragraph was incomplete. We have made an addition which includes a breif description of noradrenergic and adrenergic neurons in which VMAT2 and VGLUT2 are expressed and functions that could be disturbed in the knockout of VMAT2 in VGLUT2-neurons.

Line 254-255: “the psychostimulant amphetamine, which acts via the mesostriatal DA system.” Would be written better as “acts on/targets the mesostriatal DA system.”

 Answer: Corrected sentence.

Line 416: Up till now, the term has been dopamine-glutamate co-release, now it changes to glutamate-dopamine co-release – why?

 Answer: It was a typing error which now is corrected

Line 433: “conditioned” should be “conditional”

 Answer: Corrected.

Line 437: “disturbed” does not communicate that sensitization is attenuated in GLS1(f/f;DAT-cre) mice.

Answer: “Disturbed” changed to “attenuated”.

Line 536: “Availability of VGLUT2 in DA neurons of human individuals” -- see Root et al., Scientific Reports, 2016.

Answer: Citation included.

Line 565: The statement that “ChIs make up only about 5% of the neurons in the striatum” exaggerated the number in rodents, which is 1- 2%.

Answer: Corrected.

Lines 620 and 707: The section title “Future perspectives” is premature, since further data are discussed. It would be better to revise the title accordingly, and then add a “Conclusions” title for the last paragraph.

Answer: Agreed and corrected.  

Most of the figures are reprinted from previous publications of the authors’, but there are no statements regarding copyright permission having been obtained.

Answer: We apologize for this unclarity. All figures have been reprinted with permission from the publisher. This has now been added to the Acknowledgements.

Some of the figures, e.g. Figure 3, are not reproduced at adequate resolution.

Answer: The size of all figures has been adjusted and Figure 3 is now uploaded on the adequate quality.

Table 1: “x” is used for “no change”, but a symbol with less positive valence would be better, e.g. a dash or a horizontal double arrow.

Answer: Corrected.

Reviewer 2 Report

The authors present a scholarly, highly informative review of the roles of dopamine/glutamate co-release in models of substance abuse. The manuscript is quite comprehensive, offering readers a very well-informed survey of the literature. It is also very well-written and will undoubtedly provide a great service to the greater scientific community.

The primary comment that requires addressing is the need for better synthesis of the presented data into an overall model, particularly at the circuit level. For example, at the conclusion of the section describing data from the DAT-cre-driven VGLUT cKO and the increases in behaviors such as lever-pressing in the cKO animals (page 15), it would be helpful to present a model tying things together. Is this apparent increase/disinhibition due to decreased activation of GABA release from D2R-expressing MSNs? At a minimum, a schematic of DA/glutamate co-release and its downstream effects could be extremely informative. Similarly, can the authors speculate on the blunted responses from cKO mice to amphetamine (page 9)?

The second comment is related to some places where there is redundancy. For example, on page 11 (lines 293-296), there is apparent repetition of the findings describing blunted locomotor activity in response to amphetamine. Additionally, the intersectional approaches describing the localization of TH+/VGLUT2+ neurons to the medial VTA and their NAc projections may be better integrated earlier on in the manuscript describing the overall localization of these neurons.

Lastly, to date, it remains unclear whether TH+/VGLUT2+ neurons co-package both dopamine and glutamate into the same vesicle pools or different populations localized to different sites in the neurons. It may be worth commenting on and its functional implications.

Minor points:

It may be worth commenting more directly on work in models other than mice examining neurotransmitter co-transmission, e.g. Root et al. in primates and Aguilar et al. in Drosophila.  Small grammatical errors: line 99 (page 3): should read "can help explain some complex...", as well as line 115 (page 3), should read "putative role of VGLUTs in promoting..."

Author Response

Comments and Suggestions for Authors

The authors present a scholarly, highly informative review of the roles of dopamine/glutamate co-release in models of substance abuse. The manuscript is quite comprehensive, offering readers a very well-informed survey of the literature. It is also very well-written and will undoubtedly provide a great service to the greater scientific community.

Answer: We thank you for this summary and address your comments below.

The primary comment that requires addressing is the need for better synthesis of the presented data into an overall model, particularly at the circuit level. For example, at the conclusion of the section describing data from the DAT-cre-driven VGLUT cKO and the increases in behaviors such as lever-pressing in the cKO animals (page 15), it would be helpful to present a model tying things together. Is this apparent increase/disinhibition due to decreased activation of GABA release from D2R-expressing MSNs? At a minimum, a schematic of DA/glutamate co-release and its downstream effects could be extremely informative. Similarly, can the authors speculate on the blunted responses from cKO mice to amphetamine (page 9)?

Answer: We agree with this comment and have added a new figure showing a schematic illustration at the circuitry level (new Figure 9). We have also included an enhanced discussion regarding the amphetamine response.

The second comment is related to some places where there is redundancy. For example, on page 11 (lines 293-296), there is apparent repetition of the findings describing blunted locomotor activity in response to amphetamine.

Answer: Reformulated.

Additionally, the intersectional approaches describing the localization of TH+/VGLUT2+ neurons to the medial VTA and their NAc projections may be better integrated earlier on in the manuscript describing the overall localization of these neurons.

Answer: We agree that this could be an alternative way of presenting the study, however, we have opted to present the methodology in a more chronological order and to keep the intersectional approaches in the original position.

Lastly, to date, it remains unclear whether TH+/VGLUT2+ neurons co-package both dopamine and glutamate into the same vesicle pools or different populations localized to different sites in the neurons. It may be worth commenting on and its functional implications.

Answer: We agree to the last point, and have added a short section describing how VGLUT2 could package glutamate into pools of vesicles of DA neurons with references to relevant literature. We have also referred to other review and original articles that deal with this aspect in more detail.

Minor points:

It may be worth commenting more directly on work in models other than mice examining neurotransmitter co-transmission, e.g. Root et al. in primates and Aguilar et al. in Drosophila. Small grammatical errors: line 99 (page 3): should read "can help explain some complex...", as well as line 115 (page 3), should read "putative role of VGLUTs in promoting..."

Answer: We have added references to results originating both from drosophila and primates throughout the text but we added a clarification in the beginning of the article that the main focus in the current review is on behavioral studies on rodents. The grammatical errors have been corrected.